# AtomWorld: A Benchmark for Evaluating Spatial Reasoning in Large Language Models on Material Structures

**Taoyuze Lv** [* 1] **Alexander Chen** [* 2] **Fengyu Xie** [# 1] **Chu Wu** [1] **Jeffrey Meng** [2] **Dongzhan Zhou** [3] **Bram Hoex** [2] **Yingheng Wang** [4] **Zhicheng Zhong** [1] **Tong Xie** [# 2 5]

## Abstract

Large language models (LLMs) have shown promising potential in scientific research, enabling tasks ranging from knowledge retrieval to property prediction. Existing science benchmarks mainly focus on perceptual or knowledge-based tasks, largely ignoring the modelling tasks, a fundamental starting point for any real scientific research. For materials science, constructing and manipulating atomic structures is one of the most creative and least automated steps. In this work, we introduce AtomWorld, a benchmark designed to evaluate the abilities of LLMs on structure modifications. The benchmark includes ten fundamental actions under four widely used modelling categories, enabling verifiable evaluation metrics. We find that Claude Opus 4.6 generally performs the best. While the success rate decreases markedly with increasing modelling complexity, with particularly low success rates (below 12% for rotation) for operations involving complex spatial relations. Our results suggest that contemporary LLMs are better suited as copilots for materials structure modelling rather than fully unsupervised autonomous scientific agents. Beyond evaluation, AtomWorld also serves as a testbed and playground for developing future structure-aware models, including reinforcement learning and agentic approaches.

GitHub: AtomWorld Bench

---

[*]Equal contribution [1]Suzhou Institute for Advanced Research, University of Science and Technology of China [2]University of New South Wales, NSW, Sydney, Australia [3]Shanghai Artificial Intelligence Laboratory, Shanghai, China [4]Cornell University [5]Green Dynamics. Correspondence to: Fengyu Xie <fengyu_xie@ustc.edu.cn>, Tong Xie <tong.xie@unsw.edu.au>.

*Proceedings of the 43rd International Conference on Machine Learning*, Seoul, South Korea. PMLR 306, 2026. Copyright 2026 by the author(s).

## 1. Introduction

In many scientific and engineering studies, a core initial step is the construction of valid structural models of a system. These structures are not merely symbolic descriptions, but objects embedded in continuous space whose geometry and topology determine downstream behavior. Progress in prediction, simulation, and design therefore depends critically on the correctness and flexibility of these structural constructions. In practice, researchers frequently explore novel and combinatorially complex structures that are difficult to anticipate or encode through predefined workflows. Examples include adsorption configurations and heterostructures in materials science, molecular conformations in chemistry, and macromolecular assemblies in biology. This motivates the use of large language models (LLMs) as flexible agents capable of performing open-ended structural manipulation beyond conventional automation pipelines.

Several recent works, such as CrystaLLM, MatterGPT, and AtomGPT (Antunes et al., 2024; Chen et al., 2024; Choudhary, 2024) explore the use of LLMs to generate candidate crystal structures. However, many realistic modelling tasks cannot be solved by generating an entire structure in a single step. Instead, complex structures such as defective crystals, interfaces, or stacked heterostructures must be constructed incrementally through sequences of structural modifications. The combinatorial space of such modifications is effectively unbounded, and producing a coherent configuration requires the ability to reason over and execute structured operations step by step.

From an agentic perspective, structure modelling is not purely a perceptual or cognitive task but also an execution problem. An LLM must propose a modelling plan and faithfully apply sequences of structural operations that transform one configuration into another. This capability resembles the "motor skills" of embodied agents: the ability to carry out precise actions in a structured environment. While the perceptual abilities of LLMs have been widely evaluated through question-answer style benchmarks (e.g., LLM4MatBench (Niyongabo Rubungo et al., 2025)), much less attention has been given to evaluating their ability to perform structured manipulations. This motivates our central ques-

tion: how well can LLMs execute sequences of operations that modify atomic structures in continuous space?

In this work, we first present the AtomWorld data generation workflow, which takes input structures and produces unlimited structure-action-modified structure triples. This workflow supports benchmarking as well as future supervised or reinforcement learning (RL) of LLM agents. Using AtomWorld with a subset of structures from the Materials Project(Jain et al., 2013), we generated AtomMotor-2K, a compact test set of 2500 questions covering several fundamental action types derived from real-world structural modifications. We also designed auxiliary tests to provide additional qualitative insights. We expect AtomWorld to gain increasing relevance as LLMs improve. LLMs capability to manipulate crystal geometries is an important yet underexplored topic, and we believe the AtomWorld playground can play a foundational role in both testing and developing this in tomorrow's LLMs with rapid advancements in tool-augmented design (Hu et al., 2025), diffusion LLMs (Nie et al., 2025; Song et al., 2025), and as language-aligned video generation (Zheng et al., 2024; DeepMind, 2025) and robotics (Assran et al., 2025; Xu et al., 2026).

## 2. Related Work

**LLMs for crystallography.** To the best of our knowledge, there is currently no benchmark specifically designed to evaluate structure modification. In contrast, benchmarks for CIF generation and materials-knowledge QA do exist. LLMs have been demonstrated to hold an innate ability to generate crystal structures when pretrained on millions of CIF files (Antunes et al., 2024). This process may be further reinforced through evolutionary search frameworks (Gan et al., 2025). However, as LLMs are pattern predictors, the search space is fundamentally limited by the scope of the pretraining data. LLMs can also be instruction fine-tuned to predict crystal properties or provide general QA responses from CIF, e.g. AlchemBERT, NatureLM, Darwin 1.5, etc(Liu et al., 2025; Xia et al., 2025; Xie et al., 2025; Van Herck et al., 2025; Nate Gruver & Ulissi, 2024). In data modelling, MatText (Alampara et al., 2025a) investigates if such QA can be improved through different textual representations. Gruver et al. (Gruver et al., 2025) focused on analyzing the perception and prediction bottlenecks of LLMs when processing structured numerical data of small molecules (e.g., predicting distances or energy). Crystallography QA is benchmarked, with the most comprehensive being LLM4Mat-Bench (Niyongabo Rubungo et al., 2025), consisting of approximately 2 million composition-structure-description pairs. Crystallography benchmarks also cover multimodal LLMs, e.g. work by (Polat et al., 2025) investigates the generation of structural annotations to crystallographic images. MaCBench(Alampara et al.,

2025b) provides a more comprehensive benchmark that includes not only crystal structures but also multimodal materials and chemical experimental characterizations, showing that VLLMs still have substantial room for improvement in multimodal performance compared to text-only tasks. Tool-augmented LLMs such as OSDA Agent (Hu et al., 2025) improve structure generation through coupling computational chemistry tools to LLMs.

**Multimodal reasoning.** Approaches such as multimodal chain-of-thought (Multimodal-CoT) and visualization-of-thought (VoT) (Zhang et al., 2024; Wu et al., 2024; Yang et al., 2025) add image modalities to the reasoning trace rather than pure textual chain-of-thought. As CIF describes a 3D challenge, these results suggest that multimodal reasoning approaches can be highly applicable to improving LLM ability on CIF geometry tasks, as well as reasoning-intensive QA and structure generation/modification tasks. Approaches to multimodal representation may also be influenced from developments in video generation and robotics, where models such as Genie 3 and V-JEPA 2 (DeepMind, 2025; Assran et al., 2025) are increasingly capable of understanding real-world physics and integrating this with natural language input/output. Finally, with the training objective of diffusion LLMs (Nie et al., 2025; Song et al., 2025) to be noise reversal, they have an advantage in understanding structural text compared to autoregressive LLMs - with LLaDA (Nie et al., 2025) surpassing GPT-4o in a reversal poem completion task. This also suggests diffusion LLMs may be inherently capable of differentiating between valid and invalid modifications to CIF - important for geometric modification tasks. Advances in multimodal reasoning and diffusion indicate that LLMs may soon understand 3D CIF environments, motivating systematic benchmarking.

## 3. Playground Design: AtomWorld

### 3.1. Benchmarking Workflow

The AtomWorld benchmark consists of two stages: inference and evaluation.

During inference, each dataset sample contains an input crystal structure $s^{in}$, a natural-language action $a$, and a ground-truth structure $s^{gt}$. A prompt is constructed by combining the input structure with the action description and sent to the model. The model generates a predicted structure representation.

During evaluation, the generated output is processed through a validation pipeline. First, a CIF block is extracted from the model response. The extracted structure is then parsed and compared against the ground-truth structure. Evaluation proceeds in three steps: (1) format validation to ensure a valid CIF output, (2) composition verification to ensure

---

**Algorithm 1** AtomWorld Benchmark Evaluation

---

**Require:** Model $\mathcal{M}$, dataset $\mathcal{D}$
**Ensure:** Evaluation metrics
 1: **for** each sample $(s^{in}, a, s^{gt})$ in $\mathcal{D}$ **do**
 2:     $p \leftarrow \text{Promptor}(s^{in}, a)$
 3:     $\hat{y} \leftarrow \mathcal{M}(p)$
 4:     $\hat{c} \leftarrow \text{Extractor}(\hat{y})$
 5:     **if** $\hat{c}$ invalid **then**
 6:        record OutputFormatError
 7:        continue
 8:     **end if**
 9:     $\hat{s} \leftarrow \text{CIFParser}(\hat{c})$
10:     **if** $\hat{s}$ invalid **then**
11:        record CIFParsingError
12:        continue
13:     **end if**
14:     **if** composition mismatch **then**
15:        record AtomCountMismatch
16:        continue
17:     **end if**
18:     $(\text{correct}, \text{max\_dist}) \leftarrow \text{StructureMatch}(s^{gt}, \hat{s})$
19:     **if** not correct **then**
20:        record StructureMismatch
21:        continue
22:     **end if**
23:     record metrics
24: **end for**
25: **return** aggregated metrics

---

the correct elements and atom counts, and (3) structural matching using symmetry-aware alignment. A prediction is considered correct only if all checks succeed. We report the success rate and structural deviation statistics across the dataset.

### 3.2. AtomWorld Generator

As the core, AtomWorld generator can automatically generate benchmark data from any pre-defined structure pool and action pool. The data follows a three-part tuple: two structures of "before" and "after" states, and an action prompt describing the change - with the goal of the LLM to yield the "after" state, given the "before" state and action. A flowchart describing the workflow from data generator to benchmark is in Figure 1.

There are plenty of formats to store structure information. In Atomworld, we adopt the Crystallographic Information File (CIF) (Hall et al., 1991) format since it is one of the most well-known formats in crystallography and materials science. The CIF format contains many optional and extensible fields. Using arbitrarily mixed CIF variants would introduce additional sources of uncertainty that are not directly related

to the manipulation abilities we intend to evaluate. For this reason, we adopt the default CIF representations generated from *pymatgen* (Ong et al., 2013) and the Materials Project (MP) (Jain et al., 2013) as a standardized input format for all tasks. We leave studying how LLM performance varies across different CIF styles, as well as across other structure formats such as POSCAR and XYZ for future work.

All actions currently supported by AtomWorld are detailed in Table 1. These actions are designed to cover fundamental real-world structural modifications which researchers may perform, e.g.:

- Point defect & Doping: `change`, `remove`, `add`, `insert_between`, `swap`
- Surface generation: `delete_below`
- Structure perturbation: `move`, `move_towards`, `rotate_around`
- Supercell creation: `super_cell`

The AtomWorld playground can be also used to generate data suitable for LLM training, for instance the three-part structure of CIF-before + Action Prompt to CIF-after could feed directly into LLM pretraining. Alternatively, the same evaluation metric for AtomWorld benchmark could be used as the learning reward for reinforcement learning (RL). We leave LLM training for future work.

### 3.3. AtomMotor-2K

In principle, the AtomWorld data generator can produce an unbounded number of test cases. While AtomWorld defines a general generator for atomic structure manipulation tasks, we focus on a representative instantiation, termed **AtomMotor-2K**, to ground our analysis. AtomMotor-2K specifies a finite set of 2500 atomic actions and task templates that span ten fundamental structural operations under four widely used modelling categories, serving as a reference benchmark throughout this work.

### 3.4. Additional Probes

To support the analysis of AtomWorld, we design a set of complementary tests spanning format literacy, spatial reasoning, and property-oriented understanding. These tests play the role of breaking down AtomMotor-2K to systematically target different levels of reasoning, and also to tease the applicability of agentic CIF modification workflows through integrating tests for perceptual skills.

**Pure coordinate tests:** A stripped-down variant of AtomWorld for measuring the inherent difficulty of each geometric operation. The input becomes a set of raw coordinates like "$[[x_1, y_1, z_1], [x_2, y_2, z_2]]$". Models are then asked to

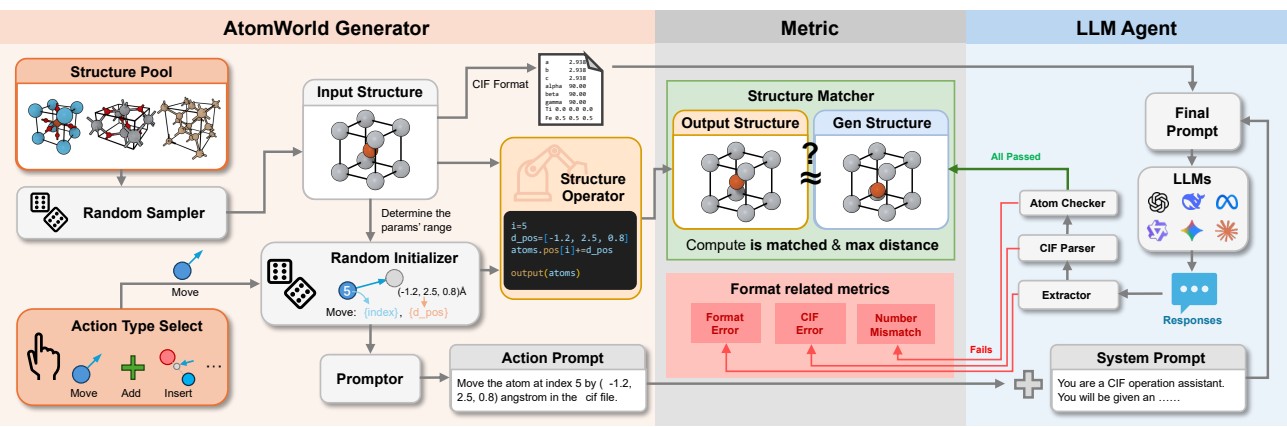

*Figure 1.* AtomWorld benchmark flowchart. The AtomWorld generator follows a structured data flow: the random sampler selects a structure from a predefined structure pool; the random initializer parametrizes the chosen action template by assigning atom indices and/or positions; the structure operator applies the instantiated action to the original structure to obtain the target structure; and the prompter generates a natural language description aligned with the action. The resulting (input structure, action prompt) pairs are then fed into the LLM agent system, whose generated structure is compared against the target structure using the `StructureMatcher` from *pymatgen* to compute the desired evaluation metric.

apply geometric operations directly on these points and return the transformed coordinates. This setting removes the complexities of CIF files and serves as a controlled test of whether the LLM can handle spatial transformations at all.

**CIF literacy tests:**

- **CIF-Repair:** Evaluates whether the model can recognize and correct corrupted or incomplete CIF files, ensuring basic robustness to noisy inputs. The CIF-Repair task is designed as the most fundamental test of CIF reading ability. The test involves CIF files with common and misleading syntax errors, such as missing tags and wrong tag names, such as "_cell_length_a" being incorrectly written as "_cell_length_x". The model is expected to correct these errors and produce a valid CIF file. A full list of corruptions is illustrated in Appendix A.6

- **CIF-Gen:** Evaluates whether the model can explicitly produce syntactically valid CIFs for simple prototype crystals (e.g., sc, fcc, bcc, perovskite), thereby examining familiarity with CIF conventions and basic materials knowledge (as opposed to the open-ended CIF generation explored by Antunes et al. (2024); Chen et al. (2024); Choudhary (2024)).

- **Chemical Competence Score (CCS):** This test assesses a model's latent chemical knowledge by evaluating its precision in distinguishing chemically accurate from inaccurate descriptions of crystal structures. While this test is a "perceptual skills" test, we use this to measure the effect that chemistry pretraining has on LLM performance in "motor skills" tasks. Following the methodology of Bran et al. (2025), the dataset was constructed by sampling 600 unique crystal structures from the MP, with

corresponding descriptions generated using Robocrystallographer (Ganose & Jain, 2019). An inaccurate dataset was then created by replacing one sentence in each original description with a sentence describing a different crystal. Because the CCS is computed from the token log-likelihoods at the model's final layer, access to these probabilities is required; this score can be calculated only for locally-run models.

**StructProp:** Highlights the deeper challenge of connecting crystal structures with their associated properties. This task is not pursued here as a systematic benchmark. Instead, we include StructProp to **underscore the importance of structural understanding as a prerequisite for materials design**, pointing toward the longer-term goal of enabling LLMs to reason about structure-property relationships. For the Struct-Prop task, a model is required to perform actions on a given structure to achieve a desired change direction in a specific property.

**Performance sensitivity tests:** Designed to test the performance under different structure conditions, including the number of atoms, Bravais lattice types, and action positions. Details can be found in Appendix B.2

## 4. Experimental Setup

### 4.1. Models Evaluated

**LLMs evaluated:** Claude Opus 4.6, Gemini 3.1 Pro Preview, Gemini 2.5 Pro, GPT-5.4, GPT-o3, GPT-o4-mini, Deepseek-V3.2 Chat, Llama-3 70B, and Qwen-3 (4B, 8B, 14B, 32B).

*Table 1.* Actions and the corresponding Action Prompt for AtomWorld.

| Action name | Action prompt |
| --- | --- |
| change | Change the atom at index {i} into {new_symbol} in the cif file. The indices of atoms are started from 0. |
| remove | Remove the atom at index {i} from the cif file. The indices of atoms are started from 0. |
| add | Add one {symbol} atom at the Cartesian coordinate {position} to the cif file. |
| move | Move the atom at index {i} by {d_pos} angstrom in the cif file. |
| move_towards | Move the atom at index {i} towards the atom at index {j} by {distance} angstrom in the cif file. |
| insert_between | Insert a {symbol} atom in the line between atoms at indices {i} and {j}, and the inserted atom must be {distance} angstrom from atom at {i} in the cif file. |
| swap | Swap the spatial positions of atoms at indices {i} and {j} in the cif file. The indices of atoms are started from 0. |
| delete_below | Delete all atoms whose z coordinate is lower than the atom at index {i} in the cif file. Excluding itself and atoms with the same z coordinate. |
| rotate_around | Rotate all surrounding atoms within {radius} angstrom of the center atom at index {i} by {angle} degree around the axis {axis} in the cif file. The rotation should following the right-hand rule. |
| super_cell | Create a supercell with the size {dim_0}×{dim_1}×{dim_2}. |

Our selection of LLMs covers frontier closed models and strong open-source baselines. We chose the Qwen-3 series to test for parameter scaling effects. We additionally conducted preliminary evaluations on several domain-specialized LLMs, including NatureLM(Xia et al., 2025) and LLaMat(Ahlawat et al., 2026). Across the first 16 tasks of each category, these models consistently failed to produce valid action trajectories, resulting in zero success rates. We therefore did not continue full-scale evaluation, as the results were already sufficiently indicative.

For tool-augmented agents, we designed a preliminary framework that enables coding with Pymatgen, using Deepseek Chat as the base model. We emphasize that the effectiveness of tool-augmented LLMs depends critically on agent design choices, including tool interfaces, action decomposition, and error recovery strategies. Here, we include a simple implementation to probe the potential benefits, rather than to provide an optimized agent. Details are in Appendix C.

### 4.2. Evaluation protocol

Our evaluation is focused on reasoning LLMs. No additional fine-tuning or reinforcement learning was performed. Inference was run with default API parameters. The prompt templates used for all tests can be found in Appendix A.4.

### 4.3. Evaluation instances

- **AtomMotor-2K.** Generated a set of 2500 AtomWorld dataframes. It contains 250 questions for the ten actions, and can be appended easily through the AtomWorld data generator. The CIFs used for "before" states are consistent across action classes as a control. A

distribution of these structures by their size is depicted in Figure 4. For the super_cell action, the output structure was specified to range from 2 to 8× the original cell size. An example of an insert_between test is illustrated in Appendix A.5.

- **Pure coordinate test.** Implemented four AtomWorld-analogous action types: move, move towards, insert_between, rotate_around. Only two points are implemented in one sample, to make the task more fundamental. For each action, 250 samples were tested on Deepseek V3, and 50 samples on Gemini 2.5 Pro. This relatively limited test sample was enough to indicate the pattern of task difficulty in AtomWorld.

- **CIF-Repair and CIF-Gen.** 22 generated samples for CIF-Repair and 20 manually-labelled samples for CIF-Gen across all LLMs used in AtomWorld. We used only a small scale of tests to isolate the LLM's understanding of CIF syntax and material structure representation from the demands of AtomWorld tasks.

- **CCS.** 600 crystal structure descriptions and their corresponding corrupted versions were generated using Robocrystallographer. As only open-source models (Llama-3 70B and Qwen3 series) were tested, the full dataset could be evaluated without the cost constraints of closed-source APIs. This dataset serves to isolate a picture of each model's latent understanding of crystal structures in natural language.

- **StructProp.** 209 manually-labelled structures are collected according to Strukturbericht type (Mehl et al., 2017). Due to the testing cost of DFT calculation pipelines, we choose 10 samples to test - for each LLM used in AtomWorld, for each property (band gap and bulk modulus). This was enough to give an indication of how

effective LLMs could be for hypothesis-driven CIF modification.

## 4.4. Metrics

**Success rate.** Used for all datasets except CCS. Defined as the number of test cases successfully pass all of the following checks divided by the total number of test cases. These errors are categorized into three hierarchical levels:

- **Wrong output format.** The LLM's response must enclose the generated structure within a predefined tag so that it can be correctly extracted from the textual output. Failure to do so constitutes an output format error.

- **Wrong structure format.** Even if the structure is successfully extracted, its file format may still be invalid or incompatible with downstream processing tools. Such cases are counted as structure format errors.

- **Mismatch of structures.** For structurally valid outputs, we compare them with the target structures using `StructureMatcher` with a site tolerance of 0.5 Å. Any generated structure whose site matching exceeds this tolerance is considered a mismatch.

**Mean maximum distance (`max_dist`).** Used for AtomWorld, pure coordinates test, and CIF-Gen. Computed only for structurally valid outputs that pass the tolerance check. For each matched pair of structures, we calculate the maximum pairwise atomic displacement after optimal alignment, and then average this value across all test cases. The `max_dist` metric is used because it is generally more significant than the RMSD value in our cases. This is because only a few or even a single atom is "moved" while others remain unchanged, making the maximum displacement a more representative indicator of the structural difference.

**CCS score.** This metric was used to evaluate whether LLMs could discern between correct and incorrect crystal structure descriptions. The underlying assumption is that models with a stronger understanding of crystal structures will assign higher likelihoods in their final layer to correct statements than to incorrect ones. Accordingly, the metric measures the separation between the distributions of mean ranks for correct and incorrect descriptions. We report this separation using Cohen's $d$ effect size, where larger values indicate a clearer distinction between the two distributions and, by extension, a stronger ability of the model to recognise correct statements based on the provided structure and its surrounding context.

**Success rate (StructProp).** The success metric for StructProp includes two additional criterion: whether the generated structure can be used in first principle calculations, and whether the modified structure fulfills the correct property change. A success rate of over 50% for a model indicates the model does better than random guessing.

## 5. Results

*Table 2.* Results for the complementary tasks. Success rate metric (%) for CIF-Repair, CIF-Gen, and StructProp datasets.

|  | CIF Repair | CIF Gen | StructProp band | StructProp elastic |
|---|---|---|---|---|
| gemini 2.5 pro | 100 | 95 | 80 | 60 |
| qwen3-32b | 59.1 | 65 | 50 | 40 |
| gpt o3 | 95.5 | 100 | 70 | 30 |
| gpt o4-mini | 90.9 | 85 | 30 | 30 |
| deepseek chat | 90.9 | 70 | 10 | 50 |
| llama3-70b | 54.6 | 0 | 50 | 10 |
| qwen3-4b | 22.7 | 0 | 40 | 10 |
| qwen3-8b | 59.1 | 20 | 40 | 40 |
| qwen3-14b | 59.1 | 55 | 40 | 20 |

### 5.1. AtomWorld: AtomMotor-2K

The main results of AtomMotor-2K are presented in Figure 2.a. We see some separation of the AtomWorld actions into easy (`change`, `remove`, `swap`, `add`), moderate (`move`, `move towards`, `insert between`) and hard difficulty (`delete below`, `rotate around`) levels based on their success rates. We also notice the mean `max_dist` metric increase for the more difficult tasks (minus tasks not requiring structural perturbations). The `max_dist` distributions for each task are shown in Appendix B.3. One interesting finding is that, the `super_cell` task cannot be well categorised into these difficulty tiers as the success rates range from Llama3-70B's 4% to 98% from GPT-o3 - it's both easy (just large-scale repetition) and difficult (requires long-context output) at the same time. The parameter scaling results in Figure 2.c and d illustrate that larger models generally achieve higher success rates and smaller displacements. However, with improvements with scale being marginal with more difficult tasks, and noting that Qwen3-32B outperformes Llama3-70B across most tasks, it suggests that architectural design and training strategies play an equally important role as parameter size. From the reasoning trace, we found that LLMs primarily approach structural tasks through explicit linear algebra and step-by-step arithmetic. Unlike humans, who have the ability to roughly visualize the positions of atoms after a modification and estimate approximate coordinates in their minds. Besides, we note that the failure cases appear largely random, with outputs ranging from unmodified input structures to partially modified or slightly perturbed atom positions. A detailed mechanistic analysis

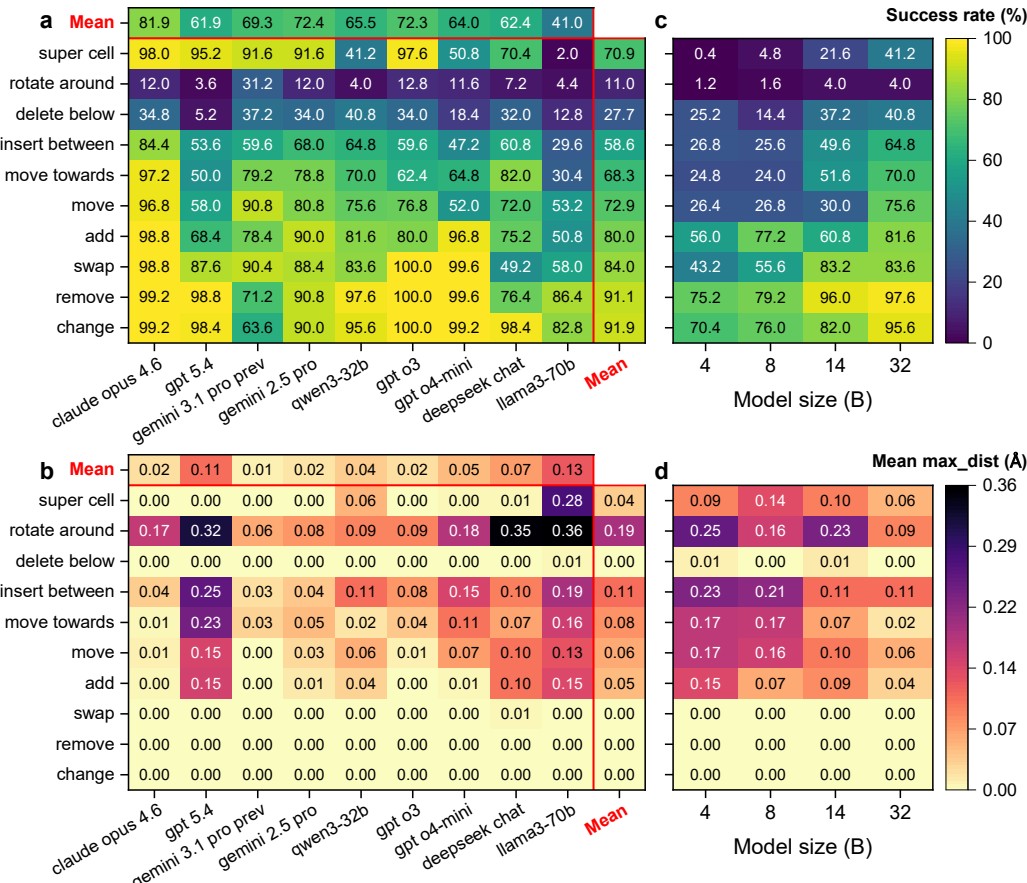

*Figure 2.* **a.** Success rate metric across AtomMotor-2K, CIF-Repair, CIF-Gen and StructProp datasets. The brighter yellow represent more positive outcomes. **b.** Mean `max_dist` metric across AtomMotor-2K and CIF-Gen datasets. **c**, **d.** Parameter scaling results on Qwen3 series. The right side are some randomly sampled structures from the tested data.

would require extensive human inspection and is left for future work.

Our evaluation of our tool-augmented LLM framework on AtomWorld tasks found noticeable gains in model performance. However, the gains are limited, particularly for more complex actions. Detailed results and comprehensive analysis can be found in Appendix C.

### 5.2. Diagnostic Results

**Pure coordinate tests.** The Results are listed in Table 3. Both models are able to reliably output a parseable output with near perfect "success rate" (errors in Deepseek V3 model at `insert_between` tasks are due to it sometimes attempting to write Python scripts instead of performing the calculation). The indicator of task difficulty is in the mean `max_dist` scores, where models performed well on `move`, `move_towards`, and `insert_between`, but found `rotate_around` significantly more difficult. The former actions involve simple, structured numerical compu-

tations (e.g., addition or weighted averaging), which LLMs tend to handle more effectively than more complex nonlinear tasks. In contrast, models often attempted to compute a rotation matrix for the `rotate_around` action and failed to apply it consistently, leading to high mean `max_dist`.

*Table 3.* Model performances on simplified point-based tasks. Mean `max_dist` is calculated by the maximum distance between generated and target points after Hungarian sort. The numbers inside the brackets represent the the ratio of readable outputs from LLMs.

| | gemini 2.5 pro 50 frames | deepseek chat 250 frames |
|---|---|---|
| Action | mean `max_dist` | |
| `move` | 0.000 | 0.000 |
| `move_towards` | 0.005 (98%) | 0.317 |
| `insert_between` | 0.005 | 0.064 (78%) |
| `rotate_around` | 16.17 (98%) | 14.06 |

**CIF-Repair.** These evaluations are presented in the main results of Table 2. Most models were able to demonstrate a strong foundational capability in understanding CIF format and errors, with success rates of over 90%. While Llama3 and Qwen3 series have success rates falling below 60%, this does not seem to limit their capability to yield higher success rates even in moderately challenging AtomWorld tasks.

**CIF-Gen.** These evaluations are presented in the main results of Table 2 and show a similar trend to CIF-Repair tasks. A closer look at the error cases in Figure 3 find that chemical compositions that define standard prototypes are generated correctly more often than non-standard compounds that crystallize in the same prototypes (e.g. NaCl vs. MgSe for rocksalt, $CaF_2$ vs. $Na_2O$ for fluorite). The fact that assymetries in training data affect LLMs in this way demonstrates that they rely more on memorization of specific examples rather than understanding the underlying structural principles. Nevertheless, Gemini 2.5 Pro and o3 were able to demonstrate this understanding with success rates of 95% and 100%, respectively.

**CCS.** The resulting scores are reported in Table 4. Similar to AtomWorld, scaling within the Qwen3 series yielded incrementally higher scores, indicating that larger models of the same architectural design acquire a more nuanced grasp of crystal structure properties from their underlying compositions. Notably, while larger Qwen models generally perform better, the Qwen3-32B model surpasses the larger Llama3-70B, mirroring the pattern observed in AtomWorld.

*Table 4.* CCS score of open-source models

| Model | CCS |
|-------|-----|
| Qwen3 4B | 0.768 |
| Qwen3 8B | 0.829 |
| Qwen3 14B | 1.061 |
| Qwen3 32B | **1.141** |
| Llama3 70b | 0.987 |

**StructProp.** These evaluations are presented in the main results of Table 2. Most LLMs were generally unable to get over 50% success rate in these tasks. With the strongest performing model Gemini 2.5 Pro achieving an average success rate of 70%, we list three examples of its reasoning trace in Table 9 in the Appendix. These examples highlight LLM knowledge of the definitions of target properties and an ability suggest plausible modification strategies, but also underlines a limited understanding of the underlying electronic structure.

In the PtS case, the model correctly identified the key driver of the band gap change as the higher energy of Se 4p com-

pared to S 3p orbitals, but stopped short of a deeper discussion of orbital overlap and covalency - Pt-S bonding is likely to be more covalent than Pt-Se, potentially leading to additional band gap narrowing. In the $Ga_2S_3$ case, the model captured the correct trend in terms of electronegativity differences and bond ionicity. The $CdAs_2$ case highlights an incorrect reasoning flow that still lead to successful completion of the task. The model mischaracterised the relative electronegativities of Cd (1.69) and Zn (1.65), attributing the improvement to enhanced ionicity - the true effect is likely linked to stronger covalent bonding due to Zn 3d-As 3p interactions.

**Performance sensitivity tests.** From the tests we found that the number of atoms (token lengths) influences performance the most, while others do not have major impacts. Details in Appendix B.2.

## 6. Discussion

The varying performance of the model on AtomWorld highlights its sensitivity to the inherent complexity of actions and the need to consider atomic coordinates and inter-atomic relationships. Actions that do not require explicit spatial reasoning or inter-atomic dependencies are easily handled (`add`, `remove`, etc.), whereas tasks involving coordinate manipulations or relational reasoning present clear challenges (`insert_between`, `move_towards`, etc.). For more complex operations or those involving many atoms, the model fails to perform reliably. This suggests that the model struggles with spatial dependencies among atoms, which limits its applicability to more complex or large-scale structural operations. This limitation is further illustrated in pure coordinate tests, where even in highly simplified environments, actions with interactions or nonlinear dependencies exhibit lower success rates, suggesting intrinsic difficulty in complex structure operations.

A key observation is that purely textual representations, while information-complete, may not be an optimal interface for structural modelling. Text encodes full structural details without loss, but requires the model to internally reconstruct spatial relationships, which can be inefficient compared to perceptual modalities. This limitation becomes more pronounced in tasks involving geometric transformations or multi-atom interactions.

Multimodal inputs, such as structural visualizations, may offer a more natural representation for such tasks. However, existing studies have shown that current vision-language models still exhibit limitations in scientific structural perception tasks (Polat et al., 2025; Alampara et al., 2025b). As a result, while multimodality is a promising direction, its practical benefits for structural modelling can remain limited at present. Nevertheless, it is likely to be an important

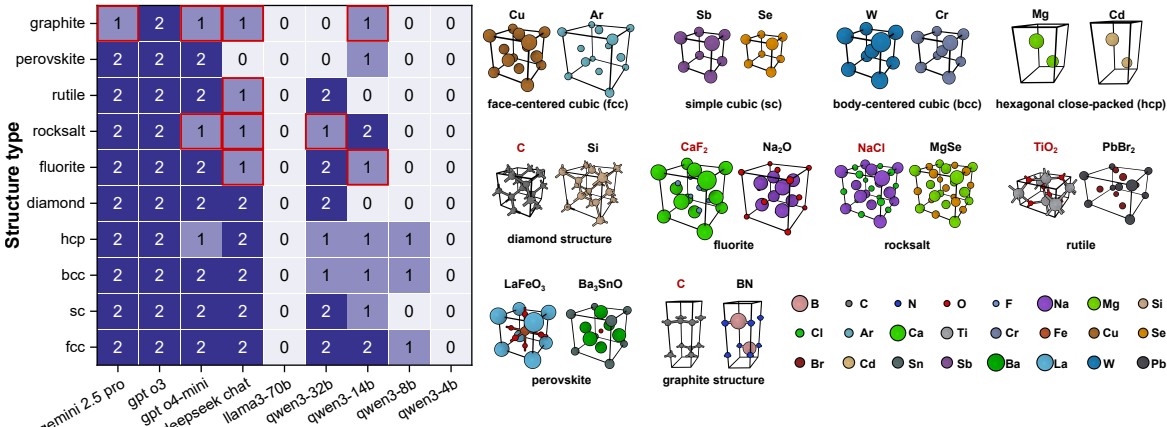

*Figure 3.* Visualised results of CIF-Gen task. The left side shows the number of correctly generated CIFs for each structure type. The squares marked in red indicate cases where the single correct generation is the standard prototype. The right side shows the specific 3D crystal structures for each type, where the chemical compositions in red represent the standard prototypes.

component of future LLM-based modelling systems.

Another limiting factor lies in the lack of effective feedback mechanisms in current modelling workflows for LLMs. Existing tools typically provide only basic checks, such as local distance validation or simple similarity metrics. This is analogous to programming environments that detect syntax errors but cannot assess semantic correctness. Consequently, even when a generated structure passes these checks, the model receives little guidance for iterative refinement. AtomWorld addresses this gap by trying to introduce verifiable evaluation metrics, which could support the development of more informative feedback systems.

Finally, tool-augmented experiments show that incorporating basic agentic workflows can improve performance, but only to a limited extent. For instance, DeepSeek's accuracy on the rotate_around operation increases from 6.8% to 18% (Appendix C). This suggests that while external tools and structured interaction can assist LLMs, they do not fundamentally resolve the underlying challenges. A more effective paradigm may require models to learn from action and feedback directly, enabling iterative reasoning over structural manipulations. We expect such action-driven learning to be broadly important for scientific tasks that involve complex interactions between operations and observations.

**Limitations** This work focuses on evaluating LLM capabilities in spatial-structure manipulation rather than full agentic modeling workflows. While we include a small set of tool-assisted experiments as preliminary exploration, a comprehensive integration of action-observation loops remains beyond the current scope. In addition, although multimodal inputs such as structural visualizations could provide complementary signals, ensuring consistent rendering and precise alignment with fine-grained structural operations is non-trivial and thus not considered yet. Finally, the current benchmark coverage is constrained by evaluation cost and task design, which limits the diversity and scale of included scenarios. We view these limitations as important directions for future work, and plan to extend the benchmark with richer multimodal inputs, broader task coverage, and more complete agentic workflows.

## 7. Conclusion

In this paper, we introduced AtomWorld, the first benchmark with verifiable, quantitative metrics for evaluating LLM motor skills in atomistic structure modelling. In general, we found that chat models took an algorithmic approach to solving the geometric tasks of our benchmark. With this approach, simpler operations such as add could be performed more consistently, whereas more spatially demanding manipulations, particularly rotations, remain highly challenging. These results imply that LLMs have limited intuitive perception of atomic structures.

Before deploying LLMs for more complex and dynamic modeling operations in real-world environments automatically, it remains necessary to conquer these verifiable structure modelling tasks and to develop modeling tools and modalities that are better aligned with LLM capabilities. Beyond benchmarking, AtomWorld can function as a playground for RL of LLM agents and a data generator for supervised fine-tuning. LLMs have traditionally struggled with spatial reasoning. Recent advances in tool-augmented models, diffusion, video generation, and language-aligned robotics suggest this limitation may soon be alleviated. We hope AtomWorld can serve as a foundational playground for evaluating and advancing LLMs' understanding of 3D CIF environments.

## Acknowledgements

T. Xie acknowledges funding from the Australian Renewable Energy Agency (ARENA) through the Australian Centre for Advanced Photovoltaics (ACAP). The views expressed herein are not necessarily the views of the Australian Government, and the Australian Government does not accept responsibility for any information or advice contained herein. T. Xie also acknowledges support from EPFL and Prof. Philippe Schwaller. T. Xie acknowledges the computational resources and support provided by the Katana computational cluster at UNSW Sydney. This work was further supported by the National Key R&D Program of China (Grant No. 2021YFA0718900), the National Natural Science Foundation of China (Grant Nos. 12374096 and 92477114), and the Jiangsu Funding Program for Excellent Postdoctoral Talent. Z. Zhong thanks the Suzhou Innovation and Entrepreneurship Leading Talent Program and the Gusu Leadership Program for their support.

## Impact Statement

This work introduces a benchmark for evaluating AI systems on materials-related reasoning and actioning tasks. By providing a systematic evaluation framework, it may help researchers better diagnose limitations of current models and guide the development of more reliable AI tools for scientific research. At the same time, benchmark performance should not be interpreted as a comprehensive measure of scientific capability, and the scope of the tasks remains limited to the current benchmark design.

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

# A. AtomWorld Setup Details

## A.1. Dataset distribution

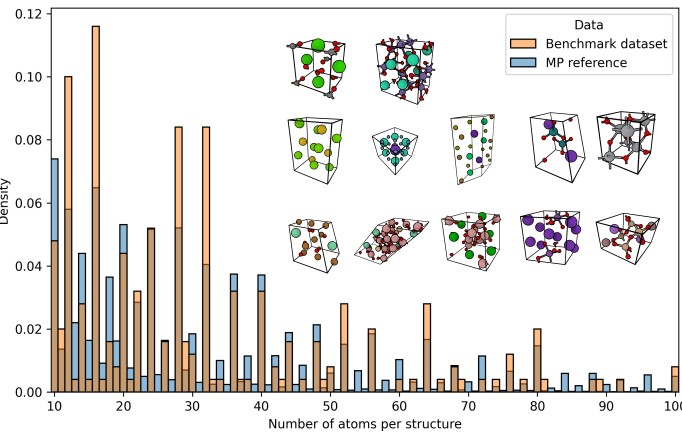

*Figure 4.* Distribution of the "before" material structures used in AtomMotor-2K (total of 250 structures) by their number of atoms. Also depicted for reference is MP's distribution and visualisations of some structures used in AtomMotor-2K.

Herein, we analyze the distributions of structures collected in our benchmark in terms of elemental composition, number of atoms, number of elements, and space groups, and compare them with those from the Materials Project. The data are shown in Figure 5, Figure 6, Figure 7, and 8, respectively. The structures in our dataset are randomly sampled from the Materials Project, with the number of atoms ranging from 10 to 100. Note that our data generator is not limited to the sampled subset; in principle, any structure file can be used for data generation, and it is capable of automatically generating an unlimited amount of benchmarking data.

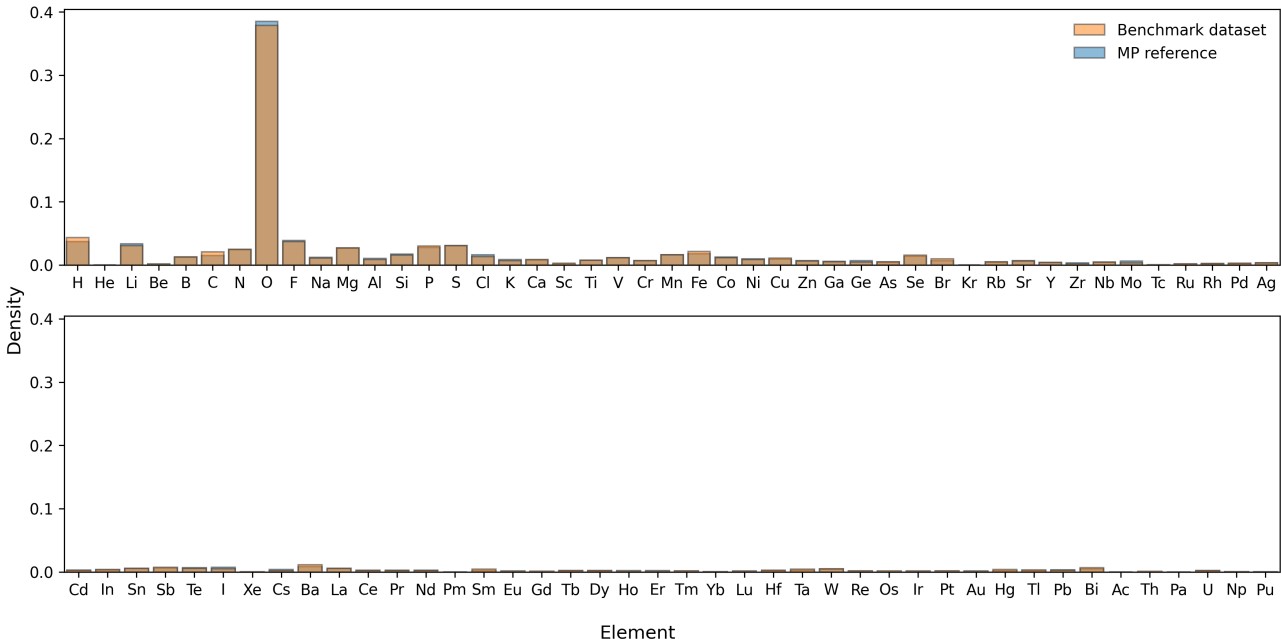

*Figure 5.* Distribution of structures by elemental composition, compared with the Materials Project. The orange bars represent the data used in the AtomWorld Bench, while the blue bars represent those of the Materials Project.

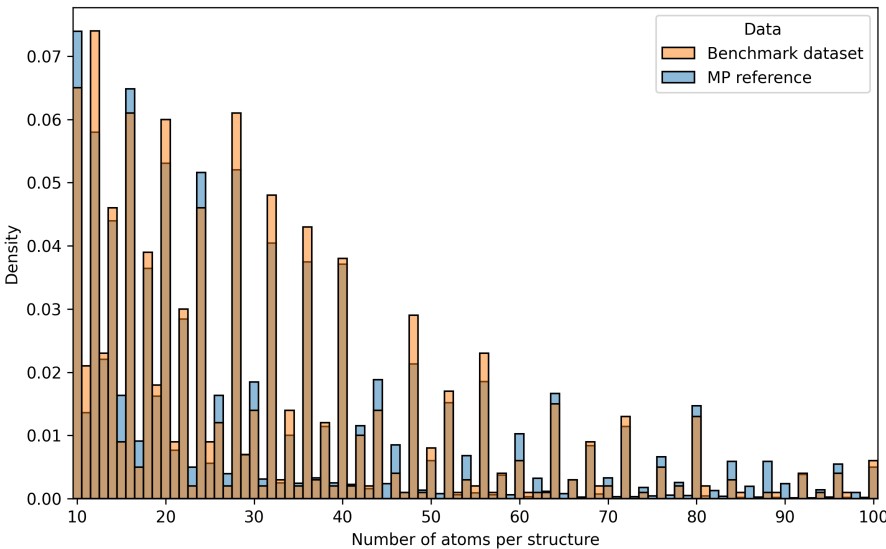

*Figure 6.* Distribution of structures by the number of atoms, ranging from 10 to 100, compared with the Materials Project.

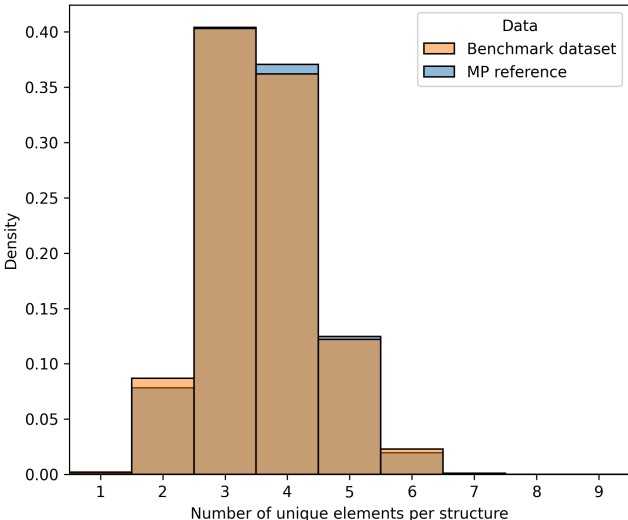

*Figure 7.* Distribution of the number of elements per structure compared with the Materials Project.

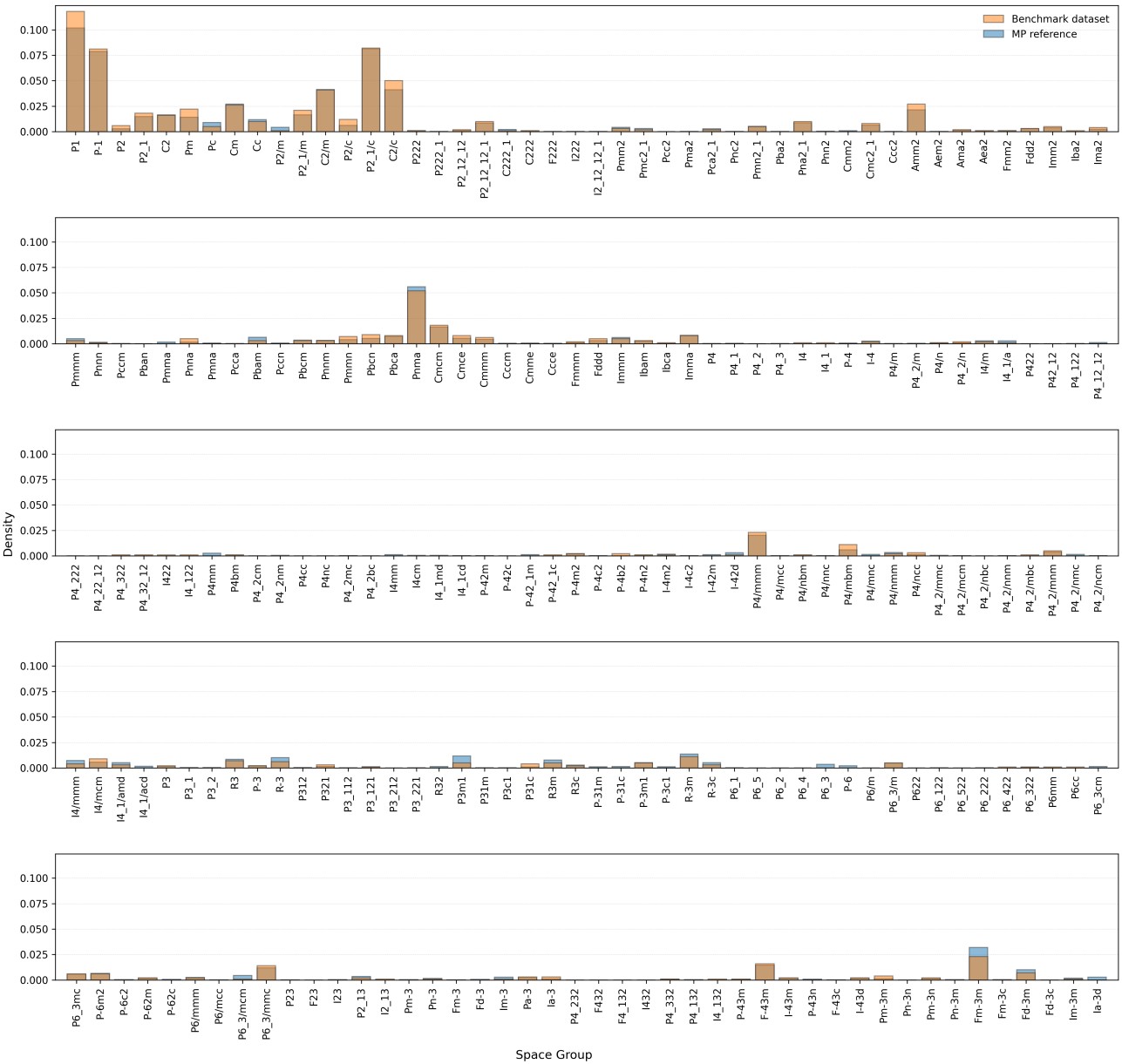

*Figure 8.* Distribution of structures by space group, compared with the Materials Project.

## A.2. Data generator parameters for each action

Each action is associated with a set of parameters that specify which atoms the action targets, as well as the magnitude and manner in which the action is applied. Our data generator takes a given structure and a specified action type as input, and produces a set of randomly initialized parameters that define the action. The generator then applies the action to the structure, producing the resulting data. Table 5 lists the parameters required for each action used in this work, along with the ranges from which the data generator randomly samples their initial values.

*Table 5.* Parameter ranges for random actions in the data generator. The input structure contains $N$ atoms, and the lattice matrix is $A = (\boldsymbol{a}_1, \boldsymbol{a}_2, \boldsymbol{a}_3)$.

| Action | Sampling ranges of parameters |
|---|---|
| change | index: $[0, N)$ 
 symbol: $\{\text{H}, \text{He}, \ldots, \text{Os}\}$ |
| remove | index: $[0, N)$ |
| add | position: $A\boldsymbol{u}$, $\boldsymbol{u} \in [0, 1)^3$ 
 symbol: $\{\text{H}, \text{He}, \ldots, \text{Os}\}$ |
| move | index: $[0, N)$ 
 displacement: $\mathcal{N}(0, \sigma^2 I_3)$, $\sigma = 2$ |
| move_towards | index1, index2: $[0, N)$, index1 $\neq$ index2 
 distance: $[0.1, 3)$ Å |
| insert_between | index1, index2: $[0, N)$, index1 $\neq$ index2 
 symbol: $\{\text{H}, \text{He}, \ldots, \text{Os}\}$ 
 distance_ratio: $[0.1, 0.9)$ |
| swap | index1, index2: $[0, N)$, index1 $\neq$ index2 
 Only between atoms with different symbols. |
| delete_below | index: $[0, N)$ 
 include_self: $\{\text{True}, \text{False}\}$ |
| rotate_around | index: $[0, N)$ 
 radius: $[1.0, 4.0)$ Å, capped by the structure size. 
 angle: $[45°, 315°)$ 
 axis: $\{\pm\hat{\boldsymbol{x}}, \pm\hat{\boldsymbol{y}}, \pm\hat{\boldsymbol{z}}\}$ |
| super_cell | size: $(a, b, c) \in \{1, 2, 3, 4\}^3$, $a \times b \times c \leq 8$, $(a, b, c) \neq (1, 1, 1)$ |

## A.3. Supported action prompts for the pure coordinate tests.

*Table 6.* Examples of actions and the corresponding action prompts for point-based tasks.

| Action name | Action prompt |
| --- | --- |
| move | Move the point at index {index} by displacement {displacement}. |
| move_towards | Move the point at index {from_index} towards the point at index {to_index} by {distance}. |
| insert_between | Insert a new point between points at indices {index1} and {index2}, {distance} units away from point {index1}. |
| rotate_around | Rotate all points by {angle_deg} degrees around the axis {axis}, with the point at index {center_index} as the center of rotation. The rotation follows the right-hand rule. |

## A.4. Full Prompt Templates

*Listing 1.* A prompt example for a specific task of AtomWorld

```
You are a CIF operation assistant. You will be given an input CIF content and an action
prompt. Your task is to apply the action described in the action prompt to the initial CIF
 content. The coordinates in the action are in Cartesian format. Return the modified CIF
content in cif format within <cif> and </cif> tags.

Please ensure the output is a valid CIF file, with correct formula, and atom positions.

Input CIF content:
{The specific CIF file is inserted here}

Action prompt: Insert Lu between atoms at indices 6 and 5 that is 4.03 angstrom from atom
6.
```

*Listing 2.* A prompt example for the pure coordinate tests

```
You are a spatial reasoning expert. You will be given an initial set of points and an
action prompt describing an operation on these points. The final modified points after
applying the action must be returned inside <answer> and </answer> tags. The format inside
 the tags must exactly match the input points format. All indices are zero-based. Please
ensure the answer inside <answer> and </answer> tags is parseable and strictly formatted.
Initial points data:
{coordinate_array},
Action prompt:
{action_prompt},
```

*Listing 3.* A prompt example for CIF-repair tasks

```
You are a CIF operation assistant. You will be given a CIF content that may be corrupted
or incomplete. Your task is to examine the CIF content and fix any issues to ensure it is
a valid CIF file. If there are missing values that cannot be repaired directly, you can
use the [VALUE_TO_BE_INSERTED] as hints to fill in the missing values. Please ensure the
output is a correct CIF file. Return the fixed CIF content within <cif> and </cif> tags.
Input CIF content:

{broken_cif}
```

*Listing 4.* A prompt example for a CIF-gen task about perovskite structure

```
You are a materials science expert. Please generate some simple and standard structures in
 the CIF format according to the requirements. You must strictly follow the CIF format
specifications. Since the symmetry-related information can be complex, please write the
CIF file with P1 symmetry. Please ensure the output is a correct CIF file. Return the
fixed CIF content within <cif> and </cif> tags.

Requirements:
```

```
Please generate a CIF file for {formula} with a {structure_type} structure, according to
the following information about the convensional cell:

- Lattice constant a: {lattice_constant_a}
- The {center_atom} atom is at the center of the octahedron formed by surrounding atoms.
```

*Listing 5.* A prompt example for StructProp tasks

```
You are a material design expert. Your task is to modify a given CIF file to achieve a
desired change in a specific material property. Please analyze the given CIF file and the
target property. Identify the key structural features and elemental composition that
influence the specified property. Propose a specific modification to the structure. This
modification must be one or a combination of the following:
 1. Element Substitution;
 2. Lattice Parameter Adjustment;
 3. Atomic Coordinate Adjustment.
 Please ensure the output is a correct CIF file. Return the modified CIF content within <
cif> and </cif> tags.
 Input CIF content:
 {The specific CIF file is inserted here}

 Your goal: modify the CIF file accordingly to {target_trend} the {target_property}.
```

## A.5. Illustrative example of the framework

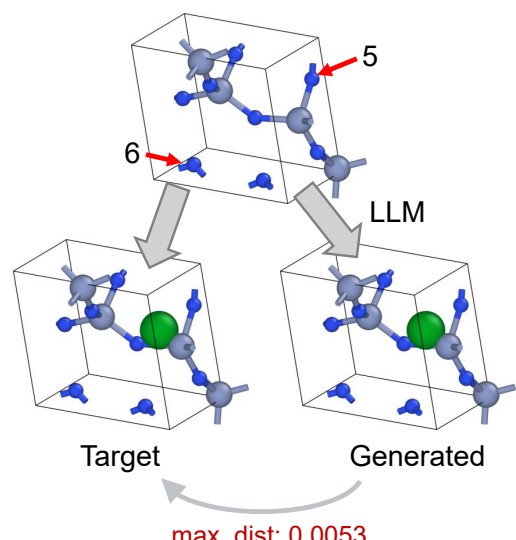

*Figure 9.* The workflow of a specific `insert_between` task.

To provide a concrete understanding of our proposed AtomWorld Bench, we present an illustrative example of its workflow. This case study focuses on a specific task: inserting a Lu atom between the fifth and the sixth atoms in the specific CIF structure. The prompt used here is listed in Appendix A.4. The workflow randomly selects the atom indices and determines the position of the atom to be inserted based on the selected atoms. Based on the initialized action, the framework gives out a target structure. The LLM will also generate a structure after processing the prompt, as shown in Figure 9. In this example, the two structures are nearly identical, with a `max_dist` of 0.0053 Å, indicating high accuracy.

Figure 10 shows a test result that is structurally mismatched to the target structure. The action prompt is: "Insert Pr between atoms at indices 12 and 13 that are 5.09 Åfrom 12 in the cif file." The LLM does appear to compute the position for inserting the atom according to the prompt during its reasoning process. However, in the final CIF file, the inserted atom is simply placed at the geometric center of the two selected atoms.

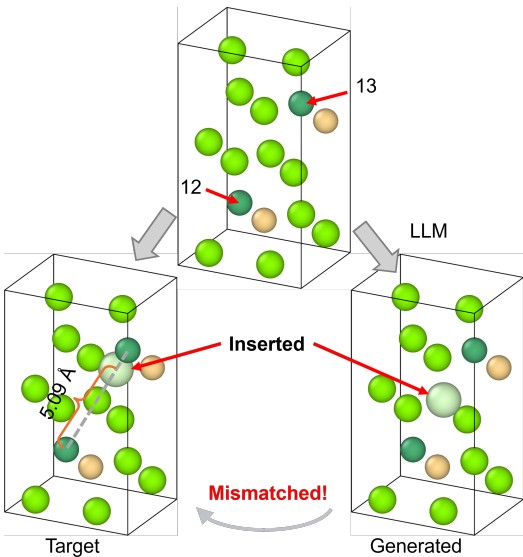

*Figure 10.* A wrong case for `insert_between` task with qwen3 32b.

## A.6. Logic on generating CIF-Repair task

To systematically evaluate LLM performance on CIF repair, we constructed a set of partially corrupted CIFs via two types of operations:

1. **Removal of essential lines:** Certain CIF fields are critical for correct structure parsing. The essential tags include:

   - `_cell_length_a`, `_cell_length_b`, `_cell_length_c`
   - `_cell_angle_alpha`, `_cell_angle_beta`, `_cell_angle_gamma`
   - `_atom_site_type_symbol`, `_atom_site_label`, `_atom_site_symmetry_multiplicity`
   - `_atom_site_fract_x`, `_atom_site_fract_y`, `_atom_site_fract_z`
   - `_atom_site_occupancy`

2. **Replacement of essential tags with misleading variants:** Instead of random typos, tags are systematically replaced with misleading but syntactically valid alternatives. Examples of mappings include:

   - Change the `a`, `b`, `c` into `x`, `y`, `z`; `u`, `v`, `w` or `i`, `j`, `k`.
   - Change the `x`, `y`, `z` into `a`, `b`, `c`; `u`, `v`, `w` or `i`, `j`, `k`.
   - Change `_atom_site` string into `_atom`.
   - Change `_cell` string into `_lattice`.
   - Change `_cell_length` and `_cell_angle` strings into `_cell`.

## A.7. DFT computation details

All density functional theory (DFT) calculations, including band gap and bulk modulus evaluations, were performed using the Vienna Ab initio Simulation Package (VASP) with the projector-augmented wave (PAW) method (Kresse & Hafner, 1993; Kresse & Furthmüller, 1996a;b; Kresse & Joubert, 1999) and the PBEsol exchange–correlation functional(Perdew et al., 2008). High-throughput workflows for both properties were automated using the atomate2 package (Ganose et al., 2025). Unless otherwise specified, calculation parameters followed the default settings in atomate2.

For the band gap calculations, a k-point mesh with a grid density of 100 Å$^{-3}$ was employed, and electronic self-consistency was converged to $10^{-5}$ eV. The band gap was extracted from the uniform k-point calculation stage. For the bulk modulus calculations, a plane-wave energy cutoff of 600 eV and a k-point grid density of 400 Å$^{-3}$ were used. Total energy and ionic relaxations were converged to $10^{-6}$ eV and 0.01 eV/Å, respectively, to balance computational cost and accuracy. In the initial relaxation stage, Gaussian smearing with $\sigma = 0.05$ eV was applied, while in the deformation stage the tetrahedron method was adopted for Brillouin zone integration.

## A.8. Fine-grained metric statistics

Here in Figure 11 and Figure 12 we show the detailed ratios for each types of error in every tasks.

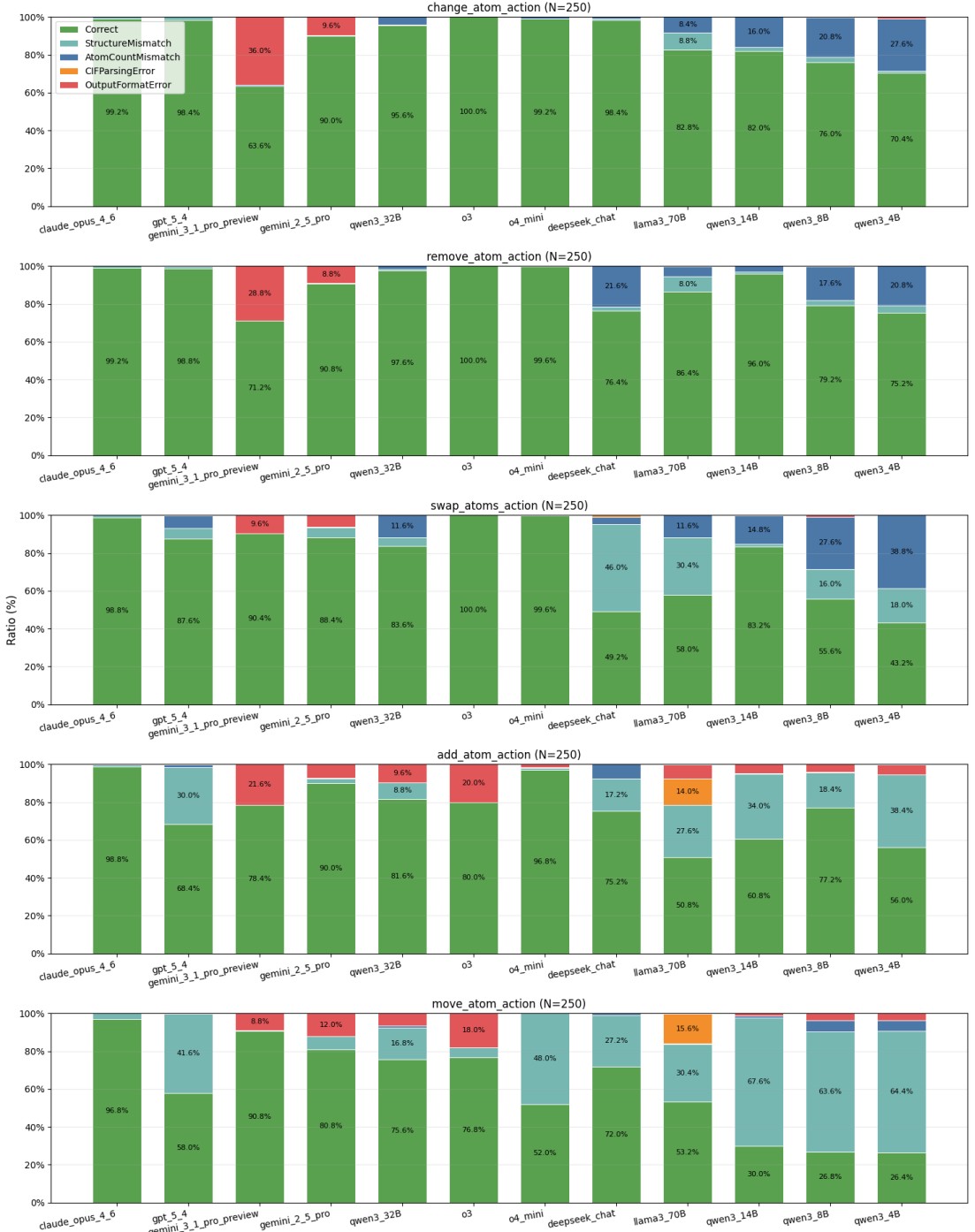

*Figure 11.* The ratio distribution of different types of error for `change`, `remove`, `swap`, `add`, and `move`, respectively.

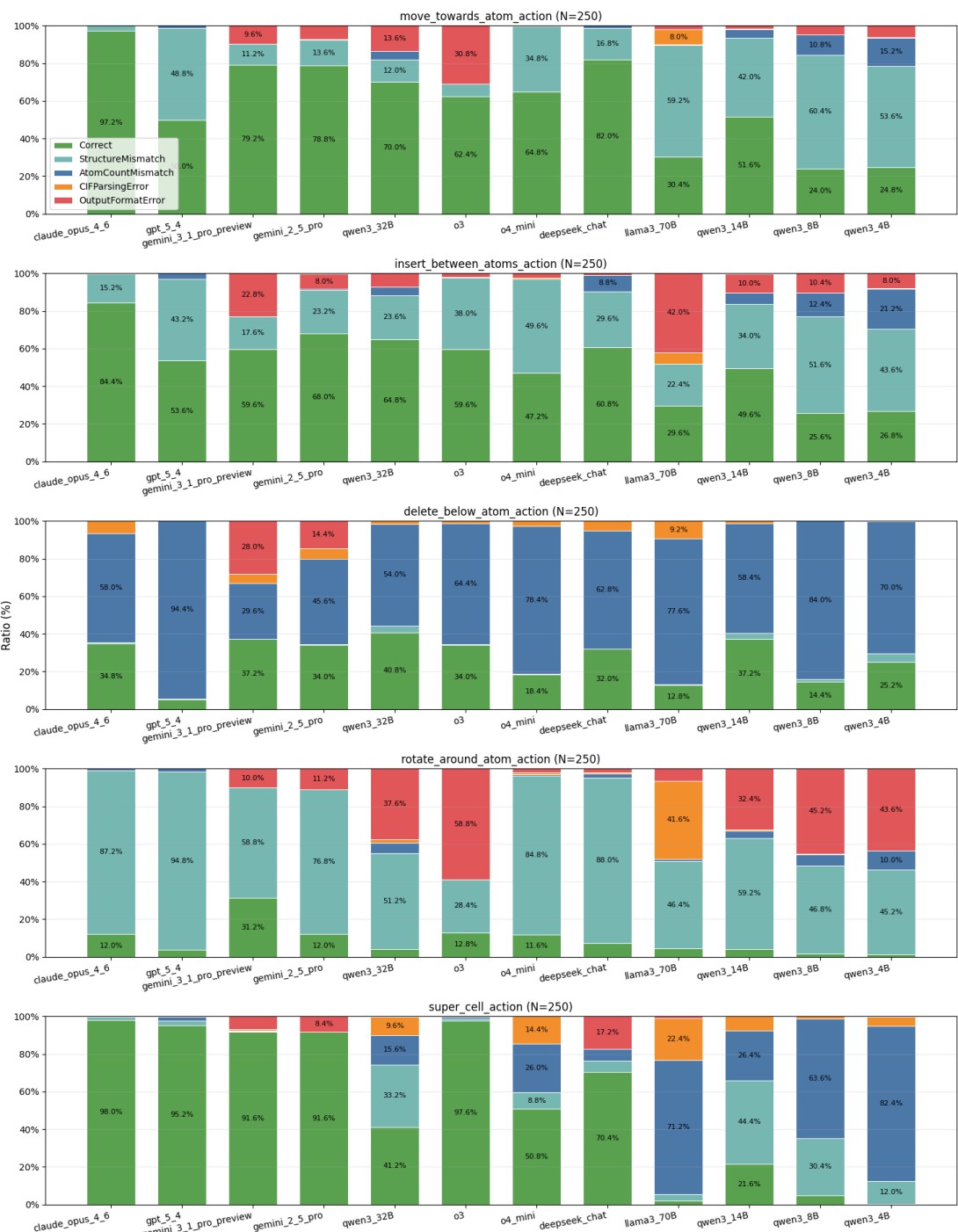

*Figure 12.* The ratio distribution of different types of error for move_towards, insert_between, delete_below, rotate_around, and super_cell, respectively.

## B. Supplementary Experimental Data

### B.1. Prompt choice and misinterpretation

During tests, we found that certain prompt formulations could cause LLMs to misinterpret spatial actions as text-level editing, specifically for the swap action (Listing 6). Prompts that focused on textual "indices" without explicitly framing

the task as a spatial transformation caused some models to attempt CIF-text rewriting instead of manipulating atomic coordinates. To verify whether this was an intrinsic ambiguity in the wording, we asked three domain experts in materials modelling to perform the same tasks using only the less explicit prompt. Both interpreted the action correctly and did not exhibit the LLM-style misinterpretation. As domain experts with long-term experience in atomic modelling, their prior exposure to structure manipulation likely provides an additional intuition for identifying the spatially intended reading, rather than a purely textual one. In contrast, LLMs appear to exhibit a default "interpretive bias" toward textual position unless the spatial nature of the task is made explicit.

Our goal, however, was not to optimize prompts, but to ensure consistent and unambiguous task interpretation. All experiments therefore use a single unified prompt set, chosen simply to remove obvious sources of misunderstanding while maintaining comparability across tasks. These prompts are likely not globally optimal but effectively prevent semantic confusion without engaging in extensive prompt tuning.

*Listing 6.* Less explicit and explicit spatial prompts for `swap` action

```
Less explicit prompt:
"Swap atoms at indices {self.index1} and {self.index2} in the cif file. The indices of
atoms are started from 0."

Success rate (Deepseek-chat):  22%
Success rate (Qwen3 32B):  50%

Explicit spatial prompt:
"Swap the spatial positions of atoms at indices {self.index1} and {self.index2} in the cif
 file. The indices of atoms are started from 0."

Success rate (Deepseek-chat):  64%
Success rate (Qwen3 32B):  98%
```

## B.2. Performance sensitivity analysis

### B.2.1. SENSITIVITY ON THE NUMBER OF ATOMS

To quantitatively investigate the influence of the number of atoms on the action success rate, we prepared test data that is as clean as possible, using a single action with consistent and simple action parameters. For the structures, we employed different supercell sizes of the same base structure. For each structure, we have tested 10 times. In this case, we chose the `insert_between` action to insert a Hydrogen atom in the middle between atoms at indices 3 and 5. We chose BiOF (mp-762304, P2$_1$/c, 12 atoms per unit cell) as the test system, and generated supercells with 12, 24, 48, 96, 144, and 216 atoms. This system was selected to provide moderate structural complexity while being representative of a three-element compound with ample data in the Materials Project.

As shown in Figure 13, the success rates of all tested models decrease as the number of atoms in the CIF increases. The larger model (Qwen3 32B) shows a slower decline compared to the smaller ones (Qwen3 4B). Beyond 200 atoms, all three models failed in every test case. This trend suggests that while larger models are more robust to increasing input size, extremely large structures still exceed the models' effective reasoning capacity.

### B.2.2. SENSITIVITY ON BRAVAIS LATTICE TYPES

We also investigated the effect of structural symmetry on the action success rate. Given that there are 230 space groups in total, analyzing all of them would be impractical. Therefore, we focused on the 14 Bravais lattices for statistical analysis. To minimize the influence of other factors, we performed random sampling from the Materials Project, selecting conventional cells containing 12 to 28 atoms, and collected 10 structures for each lattice type. As shown in Figure 14, the Bravais lattices are arranged roughly in decreasing order of symmetry, from face-centered cubic (cF) to simple primitive (aP). The success rates of the three tested models show no clear correlation with the lattice symmetry. However, the models tend to achieve slightly lower accuracy on low-symmetry systems, although a definitive conclusion would require more detailed investigation.

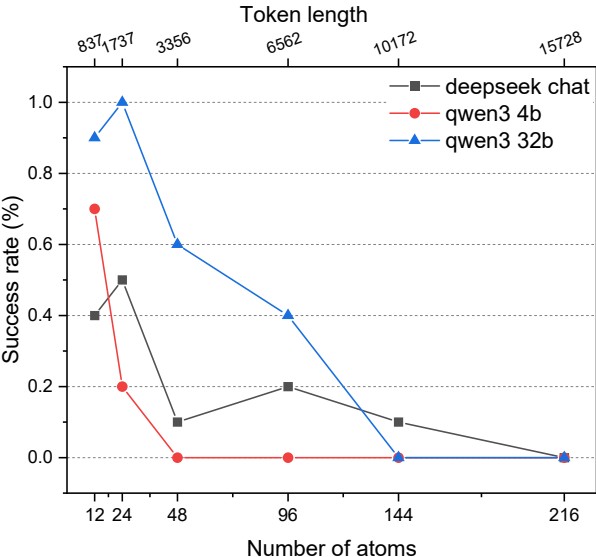

*Figure 13.* The relation between success rate and the number of atoms (directly related to token lengths).

### B.2.3. SENSITIVITY ON THE ACTION POSITIONS

We further evaluated whether the position of the target atom in the CIF sequence affects the action success rate. To this end, we prepared paired test cases using the same structure, where the target atoms appeared either early or late in the CIF atom list. All other factors were kept fixed. Specifically, we used the insert_between action with a distance_ratio of 0.45 and fixed the inserted atom type as Hydrogen. For the "early" setting, the selected atom indices were 0 and 1, whereas for the "late" setting, the last two indices in the CIF were used. The underlying structures are identical to those used in the standard insert_between tasks.

As shown in Table 7, the tested models consistently achieve higher success rates when the target atoms appear earlier in the CIF sequence. Although the difference is not large, the trend is observed across the two evaluated models. Overall, the results indicate that atom ordering has some impact on performance, but the effect remains limited under our testing conditions. To overcome this issue, we believe that the future agentic systems should use specific tools or modalities to understand the structures, then perform the actions.

*Table 7.* The success rate for different action positions.

| Action position | Deepseek-chat (%) | Qwen3 32B (%) |
|---|---|---|
| Index 0, 1 | 90 | 96 |
| Index -2, -1 | 82 | 80 |

### B.2.4. PERFORMANCE ON MOVING ALL ATOMS

We further evaluated the move_all action, which requires the model to translate the entire structure by a specified displacement vector. As summarized in Table 8, the success rates of all tested models drop substantially compared to the single-atom move tasks. In many failed cases, the generated structures show noticeable randomness in the atomic positions, indicating that the models have difficulty performing consistent global translations.

Unlike the other main metrics in AtomWorld, evaluating move_all poses an additional challenge: a rigid translation preserves the structure up to translational symmetry. As a result, the standard StructureMatcher cannot be directly applied because it intentionally factors out translational degrees of freedom. To quantify deviations, we implemented a custom metric similar to StructureMatcher but without enforcing translational invariance, comparing atoms by their index correspondence and computing the resulting coordinate deviations.

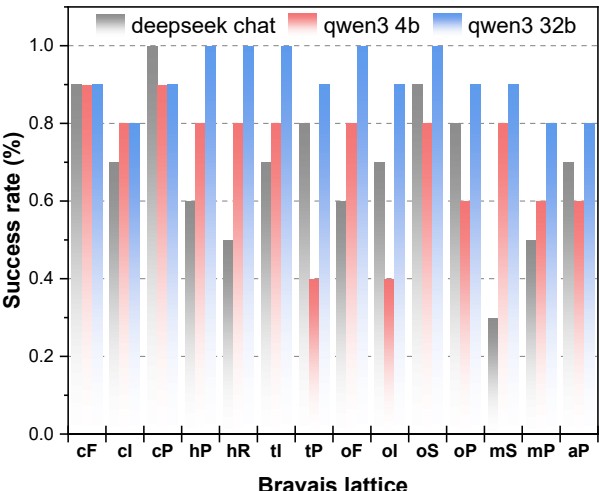

*Figure 14.* The success rate under the 14 Bravais lattice types.

*Table 8.* The metrics for the move_all action.

| Models | Success rate (%) | mean max_dist (Å) |
|---|---|---|
| Deepseek-chat | 54 | 0.1111 |
| Qwen3 4B | 12 | 0.1118 |
| Qwen3 8B | 24 | 0.0252 |
| Qwen3 14B | 40 | 0.0467 |
| Qwen3 32B | 46 | 0.0487 |

## B.3. The **max_dist** violin plots

## B.4. StructProp

*Table 9.* Excerpts from Gemini 2.5 Pro reasoning trace during successful completions of StructProp tasks. Correct and incorrect identifications are highlighted.

| Structure & Goal | Core thinking process |
| --- | --- |
| PtS (mp-288)

Band gap ↓ | ... a potent strategy is to raise the energy of the valence band maximum. This can be achieved by substituting the sulfur atoms with a heavier, less electronegative element from the same group, such as Selenium (Se). The Se 4p orbitals are higher in energy than the S 3p orbitals. |
| $Ga_2S_3$ (mp-539)

Band gap ↑ | ... its band gap is largely determined by the electronegativity difference and bond strength between the Gallium (Ga) cation and the Sulfur (S) anion. To increase the band gap, a modification that strengthens the chemical bonds and increases the material's ionicity is required. |
| $CdAs_2$ (mp-471)

Bulk modulus ↑ | ... Cd is a relatively large and electropositive element, leading to bonds with significant ionic character that are less stiff than purely covalent bonds.
... Zn is directly above Cadmium in Group 12 of the periodic table. It has a smaller atomic radius and is more electronegative, which leads to the formation of shorter, stronger, and more covalent bonds with As. |

# C. Evaluations of tool augmented LLM for AtomWorld

**System Design**   As shown in Figure 16, we adopt a code generation-based approach to accomplish structural operations. This process is divided into two steps: first, we perform RAG-based retrieval over the pymatgen library to obtain relevant APIs; second, we conduct code generation to complete the user-specified action.

**Knowledge Graph Retrieval (RAG)**   The first step of our pipeline is to retrieve relevant pymatgen APIs using RAG. We leverage the code-graph-rag project(Liu et al., 2024) to extract structured information from the codebase and build a knowledge graph in Memgraph, where nodes represent code entities such as modules, classes, methods, and fields, and edges capture relationships like inheritance and usage. The retrieval process is orchestrated by a primary LLM, implemented using Deepseek-chat, which performs task decomposition, reasoning, and tool invocation. Specifically, the translator LLM, also implemented with Deepseek-chat, is used as a tool by the primary LLM to convert natural language queries into graph queries. The output of this process is a JSON file containing relevant pymatgen APIs, which is later used to guide code generation.

**Code Generation**   Code generation is performed using Deepseek-chat, conditioned on the input CIF file, the user action prompt, and the APIs retrieved from the RAG stage. The system strictly follows the retrieved API signatures to ensure correctness and prevent hallucination. The generated Python code is then executed together with the input CIF file to produce the modified crystal structure.

As evident from Table 10, incorporating retrieval-augmented generation (RAG) and structure manipulation tools significantly improves the model's performance across the tested actions. The `remove` action, which is relatively straightforward, achieves a perfect success rate of 100%. However, more complex actions, such as `insert_between` and `rotate_around`, still present challenges. The success rate for `insert_between` is 83%, with some errors remaining, while `rotate_around` demonstrates a relatively low success rate of 18%.

These findings highlight a key insight: while the integration of RAG tools and coding ability facilitates substantial improvements in model performance, further refinements are crucial to fully address the real-world requirements of

*Table 10.* Comparison of model performances between Deepseek-chat with and without tools.

| Action | With tools | | Without tools | |
| --- | --- | --- | --- | --- |
| | Succ. rate (%) | mean `max_dist` (Å) | Succ. rate | mean `max_dist` |
| `remove` | **100.0** | 0.0000 | 84.0 | 0.0000 |
| `insert_between` | **83.0** | 0.0076 | 45.6 | 0.2004 |
| `rotate_around` | **18.0** | 0.1648 | 6.8 | 0.2561 |

structural modification tasks. Specifically, additional task-specific fine-tuning or reinforcement learning is necessary to enhance the model's robustness, particularly for more complex structural operations. Future work will focus on these aspects to ensure more reliable and scalable applications.

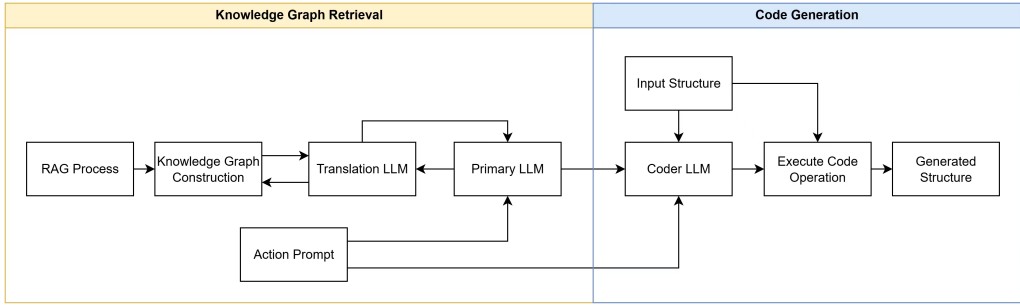

*Figure 15.* The violin plots of `max_dist` of evaluation results. The hollow squares indicate the mean values, and the hollow circles indicate the medians.

*Figure 16.* The flowchart for the code generation-based approach for the AtomWorld benchmark tests.

