# OpenReview forum: "AtomWorld: A Benchmark for Evaluating Spatial Reasoning in Large Language Models on Material Structures"
_ICML.cc/2026/Conference — ICML 2026 regular_

### Official Review · Reviewer_cdiD · 2026-03-11

**Soundness:** 3
**Presentation:** 3
**Significance:** 3
**Originality:** 2
**Overall Recommendation:** 4
**Confidence:** 4

**Summary:**

The authors propose a benchmark, AtomWorld, for testing how well LLMs understand crystal structure.
For this, they choose a functional design in which an LLM is tasked to transform structures.
Authors test different models on different transformation tasks. Especially rotations cause troubles for existing models.

**Compliance With Llm Reviewing Policy:**

Affirmed.

**Key Questions For Authors:**

I have no questions to improve my understanding.

**Limitations:**

There is no explicit section discussing limitations. As a pragmatic benchmark, this might be ok. However, one could improve clarity by adding a section clarifying that this work gives no concrete insight into what the source of the problems is and how to improve models.

**Strengths And Weaknesses:**

## Strengths
- Understanding the limitations of LLMs in dealing with crystallographic data is valuable, in principle.
- The "functional" design (by defining transformation operators) seems interesting

## Weaknesses
- It seems that the discussion of prior work is not completely fair. Limitations of modelling spatial data have been covered in MatText by Alampara et al. and https://openreview.net/forum?id=SZpygmv3G1. Multimodal understanding of crystals has been covered in https://arxiv.org/abs/2411.16955
 - It might be interesting to include different reasoning levels to measure the impact of reasoning (tokens)
- What is a bit unsatisfying is that the paper does a good job in quantifying some limitations practitioners knew about. But there is little actionable advice on what to do about them, and whether those findings translate to impact in practice (improve an architecture/training approach/...)  or at least make recommendations in this direction. This is a bit linked to the fact that there are no strong explainations for why certain operations while (besides them being "difficult").
- In principle, for a mechanistic analysis, one might need to control even variables. It is known that the token count, as well as the number of repeated tokens, can have an effect on what algorithms an LLM can implement. On the phenomenological level, for making statements like "on materials-project like structures rotations are difficult for LLMs" this is not needed. But if the authors want to dig deeper into the sources of the issues, this would be important.

## Minor points
- "The former actions could be solved with straightforward numerical calculations (e.g., addition or weighted averagng), which LLMs can handle reliably" - I am not sure if LLMs can perform such operations reliably. From my understanding, this is not the case.
- "From an agentic perspective, the structure modelling task is not solely a problem of perceptual or cognitive abilities, but one of motor skills." seems a bit far-fetched. Motor skills involve more than just spatial perception.

---

> ### Author Rebuttal · Authors · 2026-03-31
>
> Thank you for your valuable comments and here are responses for each weakness(W), question(Q), and limitation(L):
>
> **W1: The discussion of prior work is not completely fair**
> Thank you for the info, which is essential to improve our manuscript. As far as we know:
>
> - MatText focuses on building a text-based representation for structures to facilitate property prediction and generation.
> - Grover et al. focuses on analyzing the perception and prediction bottlenecks of LMs when processing structured numerical data of small molecules (energy, etc.).
> - MaCBench focuses on the multimodal understanding of materials structures.
>
> While these works are relevant, they primarily focus on static perception and prediction rather than structural manipulation. The core contribution here is the evaluation of “scientific motor skills”: the ability of an agent to sequentially and precisely edit atomic structures, which we believe is a distinct and unexplored task in this field. However, it is important to include them (MatText already cited). We are to add them in the Related Works.
>
> **W2: Might be interesting to include different reasoning levels**
> Our results already implicitly show reasoning effects — models with extended reasoning (o3, Gemini 2.5 Pro) outperform non-reasoning counterparts on geometry-heavy tasks, suggesting reasoning tokens help but don't solve the spatial limitation. Systematically varying reasoning budgets per action is an excellent direction that AtomWorld's per-operation design can support; we keep fixed inference settings here for controlled comparison.
>
> **W3: Little actionable advice**
> We highlight evidence-backed insights beyond description:
> (1) CIF literacy ≠ geometric reasoning. CIF-Repair/Gen scores predict super_cell (syntax-heavy) but not geometry-heavy tasks — refuting the assumption that CIF fine-tuning improves structural manipulation. Geometric reasoning requires fundamentally different training signals.
> (2) Tool augmentation fails where it should help most. Rotation is well-defined math with pymatgen support, yet tool-augmented DeepSeek only improves 6.8→18%. The bottleneck is geometric specification, not computation — motivating higher-level primitives and action–observation loops.
> (3) Scaling plateaus on hard tasks. Qwen3 4B→32B improves on easy tasks but stays flat on rotation (~1→4%), suggesting spatial reasoning is out-of-distribution for autoregressive training — motivating targeted RL, for which AtomWorld provides verifiable rewards.
> We will revise the Discussion to make these connections explicit.
>
> **W4: Might need to control even variables**
> We do control structure size across actions (shared CIF set), and our sensitivity analysis (Appendix B.2) shows atom count is the dominant variable while Bravais lattice type and action position have limited effect. We agree that a full mechanistic study controlling token count and repetition patterns would be valuable — AtomWorld's generator-based infrastructure is well-suited to support such investigations in future work.
>
> **W5: Not sure if LLMs can perform simple operations reliably**
> We agree the original statement was imprecise. Our intent was not to claim that LLMs perform numerical operations reliably, but rather that they often handle simple, structured computations better than more complex or less structured reasoning tasks. We have changed the original into:
>
> “…The former actions involve simple, structured numerical computations (e.g., addition or weighted averaging), which LLMs tend to handle more effectively than more complex nonlinear tasks…”
>
> **W6: Motor skills involve more than just spatial perception**
> We agree that performing the task involves more than just actions, and that cognitive abilities play an essential role.
>
> Our intention was to highlight that action-like operations serve as a foundational component: the development of operation-relevant cognitive capabilities is built upon the ability to execute and reason about such operations.
>
> We will revise the text to clarify this perspective, while still emphasizing their foundational role in structuring cognition, which is currently underexplored in this domain.
>
> **L1: No explicit section discussing limitations.**
> We agree that explicitly discussing the scope and limitations of the benchmark would improve clarity. We are going to add a limitations part in the manuscript related to:
>
> 1. This work evaluates LLM spatial-structure operation abilities rather than full agentic modeling workflows, although we include a small number of tool-assisted tests as preliminary exploration;
> 2. Multimodal inputs such as visualizations may provide additional signals for this task, but designing consistent rendering and alignment with fine-grained structural operations is non-trivial; and
> 3. The current benchmark coverage is constrained by evaluation cost and task design.
>
>  The generator-based design of AtomWorld means these extensions can be built on the same infrastructure.

---

> > ### Author Rebuttal · Reviewer_cdiD · 2026-04-03
> >
> > I remain with my weak accept. I still perceive the work as somewhat valuable, but also not a breakthrough. I also present the framing as still suboptimal as numerical and spatial reasoning limitations of LLMs have been reported before (and those are a condition for operating on things one perceives spatially).

---

> > > ### Author Response · Authors · 2026-04-03
> > >
> > > Thank you for your thoughtful assessment. However, we respectfully believe our work addresses a fundamentally different and currently underexplored aspect.
> > >
> > > Our focus is not on spatial reasoning alone, but on spatial manipulation through action. While related, these are inherently different: reasoning concerns understanding structure, whereas manipulation requires operating on it under constraints. Prior works (e.g., MatText, MaCBench, LlaMat) primarily evaluate passive understanding, whereas our benchmark targets the ability to actively construct and modify structures.
> > >
> > > We would like to further emphasize why this distinction is essential in the context of materials science and, more broadly, scientific research. **Structural modeling is a foundational component of the field**. Recent advances in bulk crystal structure generation (e.g., Nature 624, 80–85 (2023); Nature 639, 624–632 (2025)) demonstrate growing interest in generative approaches. However, even assuming ideal bulk structures, real research workflows rarely stop there. Instead, **scientists routinely construct surfaces, interfaces, and defected systems, or design entirely new configurations** based on existing structures or even created from nothing (e.g., Nat. Mater. 21, 1121–1129 (2022)).
> > >
> > > These processes cannot be reduced to fixed pipelines or simple transformations. They involve **iterative, open-ended construction under complex constraints, with a combinatorial design space that is effectively unbounded**. In practice, this is closer to a form of structured creation than to static reasoning. As a result, the ability to manipulate structures is not auxiliary, but central to enabling LLMs to meaningfully participate in computational scientific workflows.
> > >
> > > This also highlights a key implicit assumption in prior work: that scaling/improving spatial or numerical reasoning alone would translate to better actionable performance. Our results suggest this assumption does not necessarily hold.
> > >
> > > Importantly, as LLM-based agents for scientific discovery are rapidly emerging, the lack of quantitative and targeted evaluation of structural manipulation ability becomes a critical gap. Without such evaluation, it is difficult to assess or trust an agent’s capability to carry out end-to-end research workflows. This limitation cannot be addressed by conventional static benchmarks that focus on understanding alone.
> > >
> > > To the best of our knowledge, our benchmark is the first work evaluate both spatial reasoning and manipulation ability, while also **providing verifiable feedback signals** for manipulation outcomes, enabling not only evaluation but also future training through interaction.
> > >
> > > We sincerely thank you for your time and consideration. We hope this clarification better conveys the necessity and positioning of our work, and we would greatly appreciate your reconsideration of the rating. We are happy to further discuss any additional questions or concerns.

---

### Official Review · Reviewer_e1kC · 2026-03-12

**Soundness:** 3
**Presentation:** 3
**Significance:** 4
**Originality:** 2
**Overall Recommendation:** 5
**Confidence:** 4

**Summary:**

This paper introduces a novel AtomWorld benchmark aimed at evaluating the LLMs ability to understand and operate with spatial material structures. Specifically, authors propose AtomMotor-1K with 1500 templates to assess whether language models can modify crystal structures. Additionally, authors propose several higher level test templates to solve more complicated tasks. In the experiment part, authors evaluate several closed- and open-weight language models and analyse their material structure understanding capabilities.

**Compliance With Llm Reviewing Policy:**

Affirmed.

**Final Justification:**

I’m concerned that this work is useful for the LLMs for materials field and I want to maintain an “Accept” score.

**Key Questions For Authors:**

In addition to the question and suggestions stated in “Strengths And Weaknesses” section:
1. Do authors try to prompt LLMs with a few shots? Specifically, by providing examples of questions and answers before asking the benchmark question.
2. The chosen LLMs seem slightly outdated, it would be great to test the latest closed- and open-weight LLMs.
3. Also it would be great to mark in the Tables with metrics which model utilized the tools and web search to produce the answers and which is not.

**Limitations:**

Authors discuss the approach limitation in the “Discussion” section. Meanwhile, “Impact Statement” looks too shallow.

**Strengths And Weaknesses:**

The direction of material structures benchmarks is important for developing intelligent LLMs capable of understanding and operating with complex chemical structure. The idea of decomposing the evaluations of LLMs capabilities into a set of simple or moderate questions covering different aspects is not novel, but in the area of LLMs for material science is quite original. While the theoretical novelty is low, the practical part is quite strong.

The paper is written clearly and easy-to-follow.

As a moderate weakness I see the absence of experiments with LLMs fine tuning. Authors mention “This workflow supports benchmarking as well as future supervised or reinforcement learning (RL) of LLM agents”. Still, authors do not show that incorporating this knowledge through supervised fine-tuning or RL can actually improve LLMs understanding of materials. In particular, authors mention that one of subsets “the AtomWorld data generator can produce an unbounded number of test cases” and seems that that data can be utilized in massive fine tuning.

Additionally, it’s not clear how authors control the data leakage to ensure that material structures haven’t been seen through the LLM training. Do they choose for benchmarking only material structures that were published only after benchmarked LLM release?

---

> ### Author Rebuttal · Authors · 2026-03-31
>
> Thank you for your valuable comments and here are responses for each weakness(W), question(Q), and limitation(L):
>
> **W1: Absence of experiments with LLMs fine tuning**
>
> Thank you for this important suggestion. Exploring how the proposed benchmark and the generator can support supervised or RL-based training of LLMs is indeed a key direction of our ongoing work. The current paper focuses on establishing a benchmark for evaluating LLMs’ spatial and structural reasoning in materials science, and large-scale fine-tuning experiments are beyond the scope of this work.
>
> **W2: How to control the data leakage**
>
> Thank you for raising this concern. In our benchmark, the atomic structures themselves are not the primary target of evaluation. We measure the model’s ability to perform spatial operations on structures. In this sense, structures play a role similar to numbers/variables in mathematical benchmarks. The task evaluates whether the model correctly applies the specified operation rather than recalling a particular instance. Moreover, all tasks in our benchmark are programmatically generated by applying predefined operations to input structures, and such structure-operation-based data did not exist before. Therefore, even if some structures were present in pretraining data, solving the tasks still requires performing the correct structural operation rather than memorization, making potential data leakage unlikely to significantly affect the evaluation.
>
> **Q1: Do authors try to prompt LLMs with a few shots?**
>
> Thank you for the suggestion. In this work we focus on zero-shot evaluation to provide a consistent and reproducible benchmark setting across models. Few-shot prompting introduces additional prompt-design degrees of freedom and can vary substantially depending on the choice of demonstrations. Studying this systematically is an valuable direction, but is beyond the scope of the current benchmark. We plan to explore this in future work.
>
> **Q2: The chosen LLMs seem slightly outdated**
>
> Thank you for pointing this out. We have appended our test on new models . Some of the results are shown here in table format (later in fig. in the paper):
>
> ||claude opus 4.6|gpt 5.4|gemini 3.1 pro preview|
> |-|-|-|-|
> |change|99.2|98.4|63.6|
> |remove|99.2|98.8|71.2|
> |swap|98.8|87.6|90.4|
> |add|98.8|68.4|78.4|
> |mv|96.8|58|90.8|
> |mv towards|97.2|50|79.2|
> |insert between|84.4|53.6|59.6|
> |del below|34.8|5.2|37.2|
> |rot around|12|3.6|31.2|
> |super cell|98|95.2|91.6|
>
> It shows that claude opus 4.6 outperforms others in simple two-atom actions like insert between and move towards, but still cannot handle multi-atom tasks well.
>
> **Q3: Mark in the Tables with metrics which model utilized the tools and web search**
>
>  Thank you for raising this. In the current benchmark we mainly evaluate models in a controlled setting focusing on their intrinsic structural reasoning ability, therefore tool usage and web search were not enabled during evaluation (except the tool-augmented tests alone), and the reported results correspond to base model capabilities. We agree that explicitly marking tool-assisted settings would improve clarity when such capabilities are involved. Building on this work, we are actively developing an extended agentic benchmark and workflow-based evaluation where models can interact with tools and external resources. In that setting, tool usage and search capabilities will be explicitly tracked and reported.
>
> **L1:  “Impact Statement” looks too shallow.**
>
> Thank you for the suggestion. The Impact Statement followed the minimal template and was therefore intentionally concise. We agree that it can be made more informative and will expand it slightly in the revision to briefly discuss the potential benefits and limitations of the proposed benchmark. We revised:
>
> “This work introduces a benchmark for evaluating AI systems on materials-related reasoning and actioning tasks. By providing a systematic evaluation framework, it may help researchers better diagnose limitations of current models and guide the development of more reliable AI tools for scientific research. At the same time, benchmark performance should not be interpreted as a comprehensive measure of scientific capability, and the scope of the tasks remains limited to the current benchmark design.”

---

> > ### Author Rebuttal · Reviewer_e1kC · 2026-04-04
> >
> > Dear authors,
> >
> > Thank you for your responses, especially for clarifying the evaluation protocol (data leakage and tools usage). I’m still concerned that this work is useful for the LLMs for materials field and I want to maintain an “Accept” score. Please, include all stated text changes and new results to the paper.

---

### Official Review · Reviewer_Px1t · 2026-03-13

**Soundness:** 3
**Presentation:** 3
**Significance:** 3
**Originality:** 3
**Overall Recommendation:** 4
**Confidence:** 2

**Summary:**

The authors present a new material science motivated benchmark called AtomWorld. The central purpose of this benchmark is to act as a test of the ability of models to reason about material's structure and about manipulations on that structure (rotations, insertions, etc). Because of the latter, the authors motivate the impact as its value for testing the ability of models to pseudo-embodied reasoning. The agent must choose to take actions and do spatial/structural reasoning about the materials.

**Compliance With Llm Reviewing Policy:**

Affirmed.

**Key Questions For Authors:**

None

**Limitations:**

yes

**Strengths And Weaknesses:**

Strengths:
- I think this paper is well motivated and well-written. Because of the high interest in AI methods and agentic methods for material science, this is a useful contribution at a useful time.
- The connection to embodied / action-taking AI approaches is well framed and, I think, justified.
- The presented results with the benchmark are done, mostly, well and seem to give good support for the proposed benchmark.

Weaknesses:
- The models used are a little old in the AI lifecycle, but already do quite well. Is there concern that this benchmark will saturate very quickly (or could already be close to saturating with the current top models).
- I started to infer the exact wrong conclusion about Figure 2a because of the color scheme. When I first looked at it I thought white was going to be "bad" and dark "good". I can see why you did the other way, but it might be worth considering other schemes.
- My biggest concern is that content-wise, I feel like the work would be a better fit in an application journal, but the right audience is definitely the AI/ICML crowd. Perhaps there are some places to "tone down" the material science for the sake of general readers. I dont have specific suggestions so will not let this hurt the point scores below. But if the authors have some ideas of how, I think it would help the readability and potential audience.

---

> ### Author Rebuttal · Authors · 2026-03-31
>
> We appreciate your valuable comments, and here are responses for each weakness(W), question(Q), and limitation(L):
>
> **W1: Models used are a little old**
>
> Thank you for pointing this out. We have included newer models in our tests. Listed below (will be updated into figures):
>
> ||claude opus 4.6|gpt 5.4|gemini 3.1 pro preview|
> |-|-|-|-|
> |change|99.2|98.4|63.6|
> |remove|99.2|98.8|71.2|
> |swap|98.8|87.6|90.4|
> |add|98.8|68.4|78.4|
> |mv|96.8|58|90.8|
> |mv towards|97.2|50|79.2|
> |insert between|84.4|53.6|59.6|
> |del below|34.8|5.2|37.2|
> |rot around|12|3.6|31.2|
> |super cell|98|95.2|91.6|
>
> It shows that claude opus 4.6 outperforms others in simple two-atom actions like insert between and move towards, but still cannot handle multi-atom tasks well. The latest model releases are still far from saturating this benchmark.
>
> **W2: Figure 2a coloring**
>
> Thank you for the helpful suggestion. We agree that the interpretation of color schemes can vary and may lead to initial confusion. In our design, brighter colors were intended to represent more positive outcomes, which is aimed to align with a well-known materials benchmark `Matbench Discovery` and is also a commonly used convention in some visualization settings. To reduce ambiguity, we will clarify this mapping explicitly in the figure caption.
>
> **W3: "Tone down" the material science for the sake of general readers**
>
> We agree this is important for ICML positioning. Concrete changes in revision: (1) Frame the action taxonomy as general spatial reasoning primitives (local edits, relational operations, geometric transformations) rather than purely crystallographic terminology. (2) Add a paragraph connecting AtomWorld to the broader challenge of spatial reasoning and world modeling in LLMs. AtomWorld essentially probes whether LLMs possess an internal world model of 3D atomic environments — can they accurately predict the state resulting from a spatial action? This connects to the growing world model agenda in robotics and video generation (Genie 3, V-JEPA 2), grounded here in a scientifically verifiable domain. (3) Move heavy crystallography context to the Appendix where possible.
> We believe AtomWorld's design generalizes beyond crystallography to any domain requiring precise spatial manipulation — molecular design, protein engineering, CAD modeling — and will make this broader applicability explicit.

---

### Official Review · Reviewer_2k7g · 2026-03-19

**Soundness:** 2
**Presentation:** 3
**Significance:** 2
**Originality:** 3
**Overall Recommendation:** 4
**Confidence:** 3

**Summary:**

This paper introduces AtomWorld, a benchmark generator for evaluating LLMs on atomistic structure modification tasks represented in CIF format, and instantiates it as AtomMotor-1K with 1,500 test cases covering ten atomic operations across four categories. The paper evaluates several closed and open LLMs, along with a simple tool-augmented variant, using success rate and a maximum-distance metric, and reports that simpler edits such as change/remove are much easier than geometry-heavy operations such as rotation. The authors further include auxiliary probes such as pure coordinate tests, CIF repair/generation, a chemical competence score, and a preliminary structure-property task, concluding that current LLMs are better viewed as copilots than autonomous structure-modelling agents.

**Compliance With Llm Reviewing Policy:**

Affirmed.

**Final Justification:**

Thank you for the response. The author solved most of my questions.

**Key Questions For Authors:**

1.Error decomposition of main results: The paper would benefit from a detailed breakdown of failures in Figure 2a, separating at least: output-tag errors, invalid CIF syntax, and structurally valid but mismatched outputs, ideally across different actions and for the strongest models. This would clarify whether the benchmark primarily captures geometric reasoning failures or is dominated by formatting and long-sequence generation issues.

2.Justification of benchmark design choices: The rationale behind key design decisions remains unclear, particularly the uneven sample allocation (e.g., 250 vs. 50 instances per action) and the use of a fixed 0.5 Å tolerance in StructureMatcher. Additional justification or sensitivity analysis would help determine whether the reported conclusions are robust to these choices.

**Limitations:**

1.Failure in the benchmark may reflect text-generation burden rather than pure spatial reasoning.

2.Lack of deeper insights from experimental analysis.While the experiments provide useful observations, the analysis remains largely descriptive. The paper would benefit from deeper investigation into the underlying causes of model failures, offering more insightful and actionable conclusions rather than primarily reporting performance trends.

**Strengths And Weaknesses:**

Strength:

1.Novelty and problem formulation: The paper targets an underexplored evaluation gap. Instead of focusing solely on materials QA or unconditional structure generation, it emphasizes structure manipulation, which is better aligned with real scientific workflows. The distinction between “perceptual skills” and “motor skills,” although not fully formalized, effectively highlights a practically important missing dimension in current materials-LLM evaluation. Existing benchmarks largely focus on recognition, retrieval, or prediction, whereas real applications often require editing structures under geometric constraints.

2.Well-designed benchmark with diverse atomic operations: The proposed benchmark covers a reasonably broad and meaningful set of atomic editing primitives, including ten operations ranging from simple local edits to more complex relational and geometric transformations. This taxonomy provides sufficient coverage of different difficulty levels and enables the analysis of distinct failure modes, making the benchmark both practical and diagnostically useful.

3.Empirical results provide clear and informative trends: The experimental results, while not deeply explored, are directionally informative and consistent. The heatmaps in Figure 2a and 2b clearly show that models perform well on simple symbolic edits, degrade on coordinate-sensitive operations, and fail significantly on geometry-heavy transformations such as rotation. Despite some limitations in experimental depth, the central qualitative findings are coherent, plausible, and supported by the presented evidence.

Weakness:

1.Conflation of spatial reasoning and text editing: The benchmark mixes spatial reasoning with long-form CIF editing. Although framed as evaluating structural reasoning, the task also involves syntax handling, indexing, and long-context generation. As a result, failures may stem from formatting or sequence-editing issues rather than true geometric reasoning, and the current evidence does not fully disentangle these factors.

2.Imbalanced and weakly justified dataset design: The sample distribution across actions is uneven (some with 250 samples, others only 50) without clear justification. This raises concerns about the stability and reliability of per-action comparisons, especially when drawing conclusions about relative difficulty.

3.Limited depth of insight: The main findings (e.g., simple edits are easy while rotation is hard) are intuitive and expected. While useful, the analysis remains largely descriptive and does not provide sufficiently deep or actionable insights to guide future method development.

---

> ### Author Rebuttal · Authors · 2026-03-31
>
> Thank you for your valuable comments and here are responses for each weakness(W), question(Q), and limitation(L):
>
> **W1&L1: Conflation of spatial reasoning and text editing**
>
> Thank you for highlighting this. Disentangling failures from syntax-related to geometric reasoning is indeed important. Our paper already includes three lines of evidence to disentangle these factors (Section 3.3):
>
> 1. **Pointworld tests** remove all CIF syntax, reducing tasks to raw coordinates. Models still fail on rotation (mean max_dist: 16.17 Å), confirming the difficulty is geometric, not formatting-related.
> 2. **CIF-Repair/Gen** isolate format literacy. Top models score >90%, showing CIF syntax is not a bottleneck.
> 3. **Error decomposition** (see Q1): OutputErr is roughly constant across actions (6–14%), while StructMismatch scales with geometric complexity (0% for change → 76.8% for rotation). If formatting were the dominant failure mode, we would expect errors to correlate with CIF length, not geometric demands.
>
> Furthermore, CIF-Repair/Gen scores correlate with super_cell (syntax-heavy) but *not* with geometry-heavy tasks — confirming the two failure modes are separable.
>
> **W2&Q2(part):  Imbalanced and weakly justified dataset design**
>
> Originally this was due to cost considerations, but we do agree that such imbalancing would cause misleadings. Therefore we updated our manuscript to expand to consist of 250 samples for each of the 10 actions, and the results are still similar to the previous one. Please find our updated results at [**here**](https://anonymous.4open.science/r/AtomWorldBench-5632/ICML_revision/250_metric.png)
>
> **W3&L2:  Limited depth of insight**
>
> We appreciate this comment and highlight several diagnostic observations that go beyond intuitive findings.
>
> **Model diagnostics**
>
> 1. Scaling alone does not resolve spatial reasoning. While larger models improve on easier tasks(e.g., qwen3 4B→32B), performance on rotation remains nearly flat(~1→4%), suggesting these capabilities are largely out-of-distribution for standard autoregressive training rather than simply a scale limitation.
> 2. Different actions fail for different reasons. Error decomposition shows qualitatively distinct failure modes across operations (e.g., conditional filtering vs. coordinate transformation), indicating that “spatial reasoning” is not a single capability but a collection of heterogeneous skills.
> 3. Aggregate rankings hide critical gaps. Model comparisons reveal ranking inversions across tasks (e.g., o3 vs. gemini 2.5 pro on simple actions), suggesting that aggregate benchmarks can mask important capability differences.
>
> **Actionable implications**
>
> 1. CIF literacy is not geometric actioning. Recent efforts often expand CIF training data under the assumption that better familiarity with CIF syntax improves structural manipulation. Our results challenge this: CIF-repair/gen performance correlates with tasks like supercell construction but does not predict success on geometry-heavy operations. Many large models recognize CIF formats yet still fail to execute geometric actions reliably, suggesting structural manipulation requires training paradigms targeting spatial transformations.
> 2. Tool augmentation alone is insufficient. For example, rotation accuracy improves only from 6.8%→18% with tools, indicating the bottleneck lies in geometric specification rather than computation. This motivates higher-level spatial primitives and structured action-observation loops rather than raw tool access.
> 3. Need for verifiable spatial training environments. These findings motivate AtomWorld as an RL-compatible environment where spatial actions produce verifiable feedback signals, enabling targeted training of spatial reasoning capabilities.
>
> We will clarify these in the revised discussion.
>
> **Q1: Error decomposition of main results**
>
> We aim to present the per-system plots in the Appendix. Please find the figures at [**1**](https://anonymous.4open.science/r/AtomWorldBench-5632/ICML_revision/error-stack-1.png) and [**2**](https://anonymous.4open.science/r/AtomWorldBench-5632/ICML_revision/error-stack-2.png):
>
> **Q2: the set of stol = 0.5Å**
>
> The default in pymatgen is 0.3, also used in many crystal generation works (mattergen), but we have adopted a more relaxed standard to allow for a forgiving comparison rather than overly strict matching. To complement the binary match metric, we reported `mean_max_dist` to quantify the distance from predicted to true structures, providing a more continuous assessment of accuracy. The parameters do not fundamentally affect the benchmark. We have added a test on the impact of different `stol` on the benchmark success rate. As shown below: the ranking is not changed; ordinary stol will lead to lower success rate.
>
> The comparison of succ rate (%) under different stol values on insert_between action of three models
> |stol|0.3|0.5|1.0|
> |-|-|-|-|
> |gemini 2.5 pro|63.2|68.0|79.2|
> |qwen3 32B|53.2|64.8|78.8|
> |o3|51.2|59.6|78.0|

---

> > ### Author Rebuttal · Reviewer_2k7g · 2026-04-04
> >
> > Thank you for the response. The author solved most of my questions.

---

### Decision · Program_Chairs · 2026-04-30

**Decision:**

Accept (regular)

**Comment:**

This work presents a new LLM benchmark motivated by materials science. The goal is to evaluate LLMs on atomic structure modification tasks. This focus is well aligned with the needs of the scientific community.

There are some interesting outcomes here -- like the fact that scaling alone does not seem to be solving spatial reasoning for LLMs, at least in the context of atomic structure manipulation.

The primary concern is the lack of interesting machine learning insight in the paper. Some of the weaker results might be failures in text manipulation, rather than issues with understanding. However, the benchmark is novel and the application of LLMs to scientific discovery is important, and the reviewers agree the paper is well written.